# Thymopentin enhances adenoviral oncolytic therapy by regulating macrophages and CD8$^+$ T cells
Lingkai Kong[1,6], Kua Liu[1,6], Yan Liu[1,6], Huawei Cui[1,6], Peng Wang[1], Jiannan Qiu[1], Qilei Xin[2], Dan Zhou[1], Wencui Liu[2], Fangkun Zhao[2], Junnan Wu[2], Xiaosong Gu[1,2] ✉, Junhua Wu[1,2,3] ✉ & Chunping Jiang[2,4,5] ✉

## Abstract

**Background** Oncolytic viruses are cancer therapies that selectively replicate in tumors, deliver therapeutic genes, and stimulate immune responses. Combining these viruses with immune-boosting agents could enhance their effectiveness. Thymopentin (TP5), an immune-modulating peptide, may enhance the antitumour efficacy of ADV.
**Methods** We tested TP5 combined with adenovirus type 5 in tumor models, measuring tumor growth and immune changes using flow cytometry and antibody-based cell depletion. A modified oncolytic adenovirus producing TP5 (ADV-TP5) was created using gene editing and tested in mice and human immune cell-engrafted tumor models. TP5 was also combined with herpes simplex virus and vaccinia virus.
**Results** Here we show that combining adenovirus (ADV) with thymopentin (TP5) reprograms the tumor microenvironment and enhances antitumor efficacy in xenograft models. ADV + TP5 increases proinflammatory macrophages and cytotoxic CD8$^+$ T-cell infiltration while stimulating long-term immune memory. We further engineer an oncolytic adenovirus expressing TP5 (ADV-TP5), which demonstrates superior tumor suppression compared to unmodified ADV in mice and human immune cell-engrafted models. TP5 also amplifies responses when combined with other oncolytic viruses (e.g., herpes simplex virus, vaccinia virus), supporting its role as a broad-spectrum adjuvant for clinical virotherapies.
**Conclusions** TP5 strengthens oncolytic virotherapy by orchestrating macrophages and CD8$^+$ T cells to attack tumors. Its dual use, as a combination drug or engineered into viruses, offers a practical strategy to improve clinical outcomes. This approach provides a low-cost method to amplify oncolytic virus efficacy and inspires designs where single therapies mimic combination effects, advancing accessible cancer immunotherapy.

## Plain language summary

Some cancers are hard to treat because they can hide from the body's natural defenses. A promising new approach uses specially designed viruses that infect and kill cancer cells while also alerting the immune system. In this study, we combined one of these cancer-fighting viruses with an immune-boosting drug called thymopentin (TP5), which is already used in patients. When tested in mice with tumors, the virus-drug combo drew more immune cells into the tumors and improved their ability to attack cancer. To make treatment easier, we created an all-in-one virus, called ADV-TP5, that delivers TP5 directly inside tumors. This engineered virus controlled cancer growth as effectively as giving a combination of virus and TP5, in both regular mice and mice with human-like immune systems. TP5 also boosted the effect of other cancer-killing viruses. Together, these findings suggest that adding TP5 to virus-based therapies could offer a simpler, more affordable way to enhance cancer immunotherapy, and potentially help more patients benefit from this type of treatment.

Cancer remains a major health problem worldwide, although advances in medicine and scientific research have brought new hope for cancer treatment[1,2]. Traditional cancer treatments include surgery, radiotherapy, chemotherapy and targeted therapy[3]. However, while these treatments kill tumor cells, they also cause damage to normal cells, resulting in increased side effects[4–6]. In recent years, immunotherapy, as a new type of tumor treatment, has received increasing attention[7,8]. Immunotherapy relies mainly on the body's own immune system to recognize and kill tumor cells,

[1]State Key Laboratory of Pharmaceutical Biotechnology, Division of Hepatobiliary and Transplantation Surgery, Department of General Surgery Nanjing Drum Tower Hospital, Nanjing Drum Tower Hospital, the Affiliated Hospital of Medical School, Medical School, Nanjing University, Nanjing, China. [2]Jinan Microecological Biomedicine Shandong Laboratory, Jinan, China. [3]"Nanjing University-Gulou" Joint Laboratory of AI and Healthcare BigData, National Institute of Healthcare Data Science at Nanjing University, School of Life Sciences, Jiangsu Key Laboratory of Molecular Medicine, Nanjing University, Nanjing, China. [4]Department of Hepatobiliary and Pancreatic Surgery, The Second Affiliated Hospital of Fujian Medical University, Quanzhou, China. [5]Renhuai People's Hospital, Renhuai, China. [6]These authors contributed equally: Lingkai Kong, Kua Liu, Yan Liu, Huawei Cui. ✉e-mail: nervegu@ntu.edu.cn; wujunhua@nju.edu.cn; chunpingjiang@nju.edu.cn

which has excellent specificity and durability. Immune checkpoint inhibitors and adoptive cell therapy have achieved remarkable results in the treatment of a variety of malignant tumors, but there are still problems in some patients who are ineffective or relapse[9,10].

Oncolytic viruses (OVs), which can specifically infect, replicate, and lyse tumor cells while sparing normal cells, constitute a promising therapeutic approach for solid tumors[11]. OVs promote tumor regression via their ability to preferentially replicate in tumor cells, induce immunogenic cell death, and stimulate antitumor innate and adaptive immunity[12,13]. To date, adenoviruses, poxviruses, Herpes Simplex Virus Type 1 (HSV-1), poliovirus, measles virus, Newcastle disease virus (NDV), reovirus, and others have entered into early-phase clinical trials[14–16]. Moreover, because the mechanisms by which OVs eliminate tumor cells differ from those of other cancer therapies, many current treatments using OVs usually include at least one other treatment or anticancer drug, such as CTLA-4 antibodies[17], various cytokines[18,19], natural killer (NK) cells[20] and radiotherapy[21], or are genetically modified to introduce certain immunostimulatory molecules into the tumor microenvironment (TME) to increase their antitumor efficacy[22–24].

Thymopoietin II, a natural 49-amino acid polypeptide, is isolated from the thymus[25,26]. Thymopentin(TP5) is a synthetic pentapeptide corresponding to residues 32–36 of thymopoietin II (arginine-lysine-aspartic acid-valine-tyrosine, Arg-Lys-Asp-Val-Tyr)[26–28]. TP5 has been clinically used for the treatment of patients with immunodeficiency diseases, malignancies, and infections because of its immunoregulatory activity and low cytotoxicity[29,30]. Several studies have shown that TP5 can promote the differentiation of T cells by binding to T-cell-specific receptors and increasing cAMP and intracellular GMP levels, thereby inducing a series of intracellular reactions and regulating immune functions[31,32]. The clinically approved immunomodulator TP5 may be a suitable agent for promoting the antitumor efficacy of OVs.

In this study, we first test the antitumor effects of combining adenovirus (ADV) with thymopentin (TP5) in a variety of mouse models. Our results show that ADV combined with TP5 exhibits superior antitumor efficacy compared with other treatments and even leads to complete tumor regression and tumor-specific immune memory in the H22 tumor model, which may be related to the fact that the combination can effectively increase the number of cytotoxic CD8$^+$ T cells and proinflammatory macrophages in the TME. Further experiments demonstrate that the depletion of macrophages and CD8$^+$ T cells impairs the efficacy of the ADV and TP5 combination treatment. Moreover, we construct a replication-competent adenovirus expressing TP5, which harnesses all the advantages of the aforementioned combination therapy components. We also test the antitumor efficacy in murine xenograft tumor models, PBMC-humanized CDX models and PDX models. It is worth noting that the promoting effect of TP5 on oncolytic virus therapy is not limited to adenoviruses, it can also combine with various oncolytic viruses (such as HSV and VV) to produce an effective antitumor immune response. Given that TP5 has been clinically approved as an immunomodulator in many countries worldwide, our study provides a highly promising OV combination treatment strategy for clinical translation, along with a promising recombinant oncolytic adenovirus for cancer therapy.

## Methods

### Mice
Four- to eight-week-old female mice were used for the experiments to control for potential sex-based differences in immune responses and tumor growth, thereby enhancing experimental consistency. Wild-type (WT) BALB/c mice and immunodeficient BALB/c-nude mice were purchased from Nanjing University Model Animal Institute. All the mice were housed under specific pathogen-free conditions at ~18–24 °C with free access to water and food and maintained under a 12-h/12-h light/dark cycle. Animal experiments were performed in accordance with the ARRIVE guidelines. All animal experiments were approved by the Ethics Committee of The Affiliated Drum Tower Hospital, Medical School of Nanjing University.

### Cell lines
The mouse breast cancer cell line 4T1 (ATCC, CRL-2539), the mouse colorectal cancer cell line CT26 (ATCC, CRL-2638) and the human embryonic kidney cell line 293 T (ATCC, CRL-3216) were cultured in DMEM supplemented with 10% FBS, 100 U/mL penicillin, 0.1 mg/mL streptomycin (Gibco), and the mouse hepatocellular carcinoma (HCC) cell line H22 (CCTCC, IM-M003) was cultured in RPMI 1640 medium (Life Technologies). All the cells were incubated at 37 °C with 5% $CO_2$.

### Human samples
Studies involving human participants and samples were conducted in accordance with the principles of the Declaration of Helsinki. Human tissue collection and subsequent studies were approved by the ethics committee of Nanjing Drum Tower Hospital, the informed consent was obtained from all participants (Approval number: 2016-05-17).

### Construction of the recombinant virus
In a previous study[33], an adenovirus shuttle plasmid encoding the TP5 domain of thymopentin II was purchased from Sino Biological. For the secretion and detection of TP5, the IL-2 signal peptide (MYRMQLLS-CIALSLALVTNS) was designed upstream of the TP5 sequence, and the His-tag (HHHHHH) was designed downstream of the TP5 sequence. After digestion with the restriction enzyme PacI, the recombinant adenovirus was generated and amplified in 293T cells. The recombinant adenoviruses were purified via sucrose gradient ultracentrifugation and titrated via the Adeno-X™ rapid titer kit according to the manufacturer's instructions (Cat. No. 632250, Clontech).

HSV and VV viruses (Cat#VR-1540-ATC) expressing EGFP were purchased from Wuhan Binhui Biotechnology.

### Western blot analysis
First, 4T1 cells were seeded into 6-well plates and infected with ADV or ADV-TP5 at an MOI of 2 for 48 h. Supernatants from infected cells were harvested and mixed with loading buffer. The following experimental steps of western blotting were performed according to previously described procedures[34]. The antibodies used were anti-6x-His tag monoclonal (Invitrogen, 4E3D10H2/E3, 1:1000) and goat anti-mouse IgG (H + L) (Invitrogen, 31430, 1:20,000).

### Oncolytic virus treatment and replication
4T1, CT26, and H22 cells ($2 \times 10^3$) were seeded into 96-well plates and cultured overnight prior to treatment with different adenoviruses for 48 h. Cell viability was evaluated via a CCK-8 assay. 4T1, CT26, and H22 cells ($1 \times 10^5$) were seeded into 24-well plates and infected with different adenoviruses (MOI 2 or 4). The medium containing the viruses was removed, and fresh medium was added 2 h after infection. The cells were harvested at several time points as indicated (6, 24, 48, 72, and 96 h after infection). Total RNA was subsequently isolated with TRIzol (Invitrogen) and used to generate cDNA with a HiScript Reverse Transcriptase Reagent Kit (Vanzyme Biotech, China). Viral replication was calculated via the comparative threshold cycle of E1A and normalized to GAPDH expression.

### qPCR
Total cellular RNA was collected and then reverse transcribed to cDNA, as described in previous viral replication experiments. mRNA expression levels were analyzed via SYBR Green PCR master mix (Vanzyme Biotech, China) and an ABI QuantStudio 5 real-time PCR system (Thermo Fisher Scientific) and normalized to GAPDH expression. The following primer sequences were used for PCR. E1A: 5-CCTTCTAACACACCTCCTGAGATACA-3, 3-CAGGCTCGTTAAGCAAGTCCTC-5; GAPDH: 5-AGGTCGGTGT-GAACGGATTTG-3, 3-TGTAGACCATGTAGTTGAGGTCA-5.

### Flow cytometry
The samples were run on a FACSCaliber cytometer (BD) and Beckman Coulter Cytoflex S and analyzed with FlowJo 10. For the immune cells in the

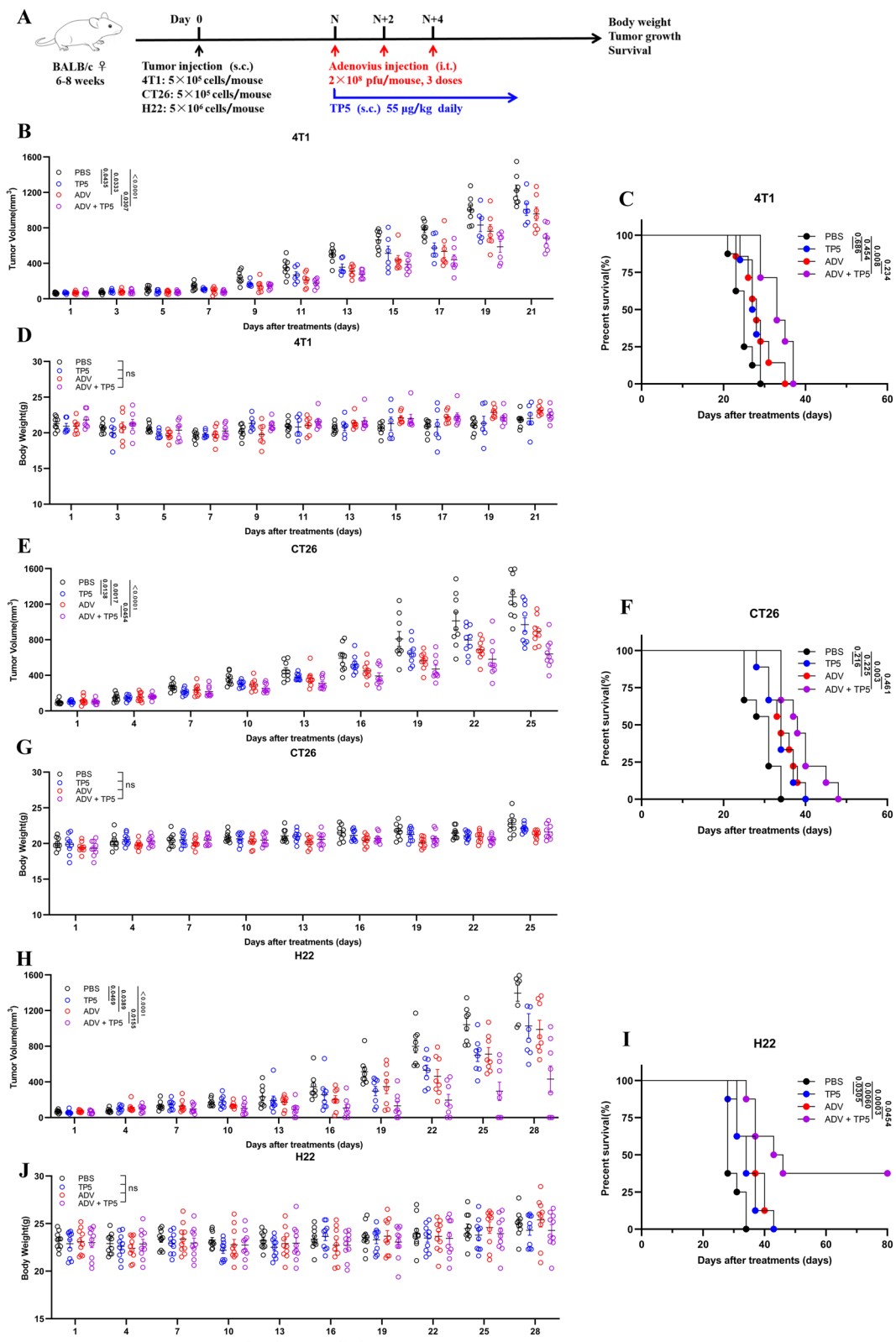

TME and in the spleen, tissues from the mice were collected and filtered to generate single-cell suspensions by digestion with collagenase IV (50 μg/mL) for 1 h at 37 °C. For the peritoneal macrophages, the cells were collected from the peritoneal lavage fluid and centrifuged at 1000 rpm for 5 min. For the polarization of macrophages in vitro, cells were collected at the indicated time points after different treatments. The harvested cells were stained with different antibodies. Fluorescent antibodies recognizing murine CD45-APC, CD3-FITC, CD4-PE-Cy7, CD8-PerCP-Cy5.5, IFN-γ-PE, FoxP3-PE, CD11b-FITC, F4/80-PE-Cy7, CD86-PE, CD206-PerCP-Cy5.5, CD69-PE-Cy7, CD49b-PE, CD25-APC-Cy7, and CD11c-PE-Cy7 were used in this assay and were acquired from Biolegend. Cell viability was assessed using DAPI to identify dead cells. Fluorescence-minus-one (FMO)

**Fig. 1 | TP5 increases the antitumor efficacy of oncolytic adenoviruses in multiple tumor models. A** Experimental protocol for the establishment of tumor-bearing mice and their subsequent treatment. BALB/c mice were subcutaneously injected with tumor cells. When the tumor volume was ~50–100 mm³, the mice were randomly divided into different groups and treated with PBS or ADV three times every 2 days. TP5 was injected subcutaneously into the peritumoral site once daily after the first dose of ADV or PBS. **B–D** 4T1 model (*n* = 8 mice for PBS; *n* = 6 mice for TP5 and ADV groups; *n* = 7 for ADV + TP5 group). **B** Tumor growth data are presented as the mean ± SD. **C** Kaplan–Meier survival curves for all groups. **D** Body weight data are presented as the mean ± SD. **E–G** CT26 model (*n* = 9 mice per group). **E** Tumor growth data are presented as the mean ± SD. **F** Kaplan–Meier survival curves for all groups. **G** Body weight data are presented as the mean ± SD. **H–J** H22 model (*n* = 8 mice per group). **H** Tumor growth data are presented as the mean ± SD. **I** Kaplan–Meier survival curves for all groups. **J** Body weight data are presented as mean ± SD. Data are presented as mean ± SD, and *P* values were calculated using the two-way ANOVA with Geisser-Greenhouse correction (**B, D, E, G, H, J**), log-rank test in (**C, F, I**). ns, no significant difference.

controls were used for accurate gating in flow cytometry. The following cell gating strategies were used: lymphocytes (FSC-H and SSC-H), single cells (SSC-A and SSC-H), CD45+ cells (CD45+ gated single cells), CD3+ T cells (CD3+ gated CD45+ cells), CD4+ T cells (CD4+ gated CD45+ cells), CD8+ T cells (CD8+ gated CD3+ cells), activated CD8+ T cells (CD25+CD8+ or CD69+CD8+ or IFN-γ+CD8+ gated CD45+ cells), Tregs (CD25+FoxP3+ gated CD4+ T cells), macrophages (CD11b+F4/80+ gated CD45+ cells), "M1-like" macrophages (CD86+ gated macrophages), "M2-like" macrophages (CD206+ gated macrophages), DCs (CD11c+CD86+ gated CD45+ cells), and NK cells (CD3-CD49b+ gated CD45+ cells), lymphoid and myeloid cells were stained with antibodies separately and then analysed by flow cytometry.

For apoptosis analysis, tumor cells were incubated with viruses (MOI of 10 or 20). After 24 h, the cells were collected and stained with Annexin-V/PI (Elabscience) for 20 min.

### In vivo experiment

For the solid tumor model, exponentially growing 4T1, CT26, or H22 cells were harvested and then injected subcutaneously into the flanks of the mice. The tumor volume was calculated via the formula length × width² × 0.5. Once the tumor volume reached ~50–100 mm², the mice were randomly divided into different groups. Then, 0.1 mL of PBS, ADV ($2 \times 10^8$ PFUs), HSV ($1 \times 10^7$ PFUs), VV ($1 \times 10^7$ PFUs) or ADV-TP5 ($2 \times 10^8$ PFUs) was injected intratumorally every other day for a total of three times, and TP5 was injected subcutaneously once daily. The tumor volume was measured three times per week during the treatment. For survival experiments, the mice were sacrificed when the tumor volume reached ≥1500 mm³.

For the rechallenge model, the cured mice administered immunotherapies were injected subcutaneously with $5 \times 10^6$ H22 cells on day 0. For the bilateral tumor-bearing model, the cured mice with immunological memory were rechallenged with $5 \times 10^6$ H22 cells, and the opposing flank was injected subcutaneously with $5 \times 10^5$ 4T1 cells on day 80 after the first rechallenge. Naive mice were used as controls. The mice were sacrificed when the tumor volume reached ≥1500 mm³.

For macrophage and CD8+ T-cell depletion, BALB/c mice were intraperitoneally injected with 500 µg of anti-CD8a (Bioxcell, West Lebanon, NH, USA) or anti-CSF1R (Bioxcell, West Lebanon, NH, USA) antibodies every other day for a total of three injections.

For the PBMC-humanized CDX model, exponentially growing MDA-MB-231 and/or HepG2 cells were harvested and then injected subcutaneously into the flanks of NCG mice (GemPharmatech, NOD/ShiLtJGpt-Prkdc^em26Cd52^Il2rg^em26Cd22^/Gpt) on day 0. PBMCs (obtained from human peripheral blood) were inoculated into the mice on day 1. When the tumor volume reached ~50–100 mm³, the mice were randomly divided into different groups. Then, 0.1 mL of PBS, ADV or ADV-TP5 ($2 \times 10^8$ PFUs) was injected intratumorally every other day for a total of three times, and TP5 was injected subcutaneously once daily into the peritumoral site. The tumor volume was measured three times per week during the treatment. For survival experiments, the mice were sacrificed when the tumor volume reached ≥1500 mm³.

For the PBMC-humanized PDX model, TNBC tumor cells were resuspended in 50% Matrigel (Invitrogen, A1413202) and injected into the fourth mammary fat pads of NCG mice on day 0. PBMCs (obtained from human peripheral blood) were inoculated into the mice on day 1. When the tumor volume reached ~50–100 mm³, the mice were randomly divided into different groups. Then, 0.1 mL of PBS, ADV or ADV-TP5 ($2 \times 10^8$ PFUs) was injected intratumorally every other day for a total of three times, and TP5 was injected subcutaneously once daily into the peritumoral site. The tumor volume was measured every other day.

For generation of the PDX model, the excised breast tumors of patients diagnosed with triple-negative breast cancer (TNBC) were maintained on ice and brought to the laboratory within one hour, after which they were homogenized and digested to generate single-cell suspensions and/or organoids. The cells were filtered through a 70-µm sterile filter. A total of $5 \times 10^6$ viable tumor cells were resuspended in 50% volume Matrigel and injected into the fourth mammary fat pads of NCG mice. When the tumors reached ~1000 mm³, they were harvested and dissociated into single cells as previously described. The initial tumor (which reached a volume of 1000 mm³ in NCG mice) was termed 'passage 0' (P0), and passages continued to be tracked with each generation. The oncolytic adenovirus (ADV + TP5 and ADV-TP5) was used to treat the PDX model mice at passage 3.

### Statistics and reproducibility

Statistical analyses were conducted with GraphPad Prism 9. Data are expressed as mean ± SD. Group differences were evaluated by two-tailed unpaired *t* test with Welch's correction (for two groups) or one-way/two-way ANOVA (for multiple groups). Survival analysis was performed using the Kaplan-Meier method, with comparisons made by the log-rank test. For in vivo studies, mice were randomly allocated to treatment groups using computer-generated random sequences once tumors reached ~50–100 mm³. While the treatments (e.g., virus injection) could not be fully blinded to the personnel administering them, endpoint measurements (tumor volume, flow cytometry) and data analysis were performed by investigators blinded to group allocation. No mice were excluded from the analyses. Sample size was not predetermined using statistical methods. In all tests, *P* value < 0.05 was defined as statistically significant. The exact sample size (*n*) for each experiment is provided in the corresponding figure legend.

### Results

#### TP5 effectively enhances the antitumor effect of ADV in a variety of solid tumor models

To evaluate the antineoplastic effects of ADV and TP5 and the related mechanisms, multiple tumor mouse models were established, and tumor-bearing mice were treated with ADV and/or TP5 every 2 days for a total of three times (Fig. 1A).

4T1 tumors generally lack tumor-infiltrating lymphocytes (TILs) and are unresponsive to immunotherapies[35]. In 4T1 tumor-bearing mice, ADV combined with TP5 significantly delayed tumor growth (Fig. 1B) and effectively prolonged survival (Fig. 1C). We also found that the body weights of the tumor-bearing mice in each group did not significantly differ, indicating that ADV and TP5 had low toxicity in 4T1 tumor-bearing mice (Fig. 1D).

The antitumor effect of the combination therapy was also confirmed in a CT26 tumor model. Compared with PBS, ADV alone and/or TP5 alone had a positive therapeutic effect on CT26 tumor-bearing mice (Fig. 1E–G). Similarly, ADV combined with TP5 significantly delayed tumor progression (Fig. 1E–G). Additionally, we also found that ADV combined with TP5

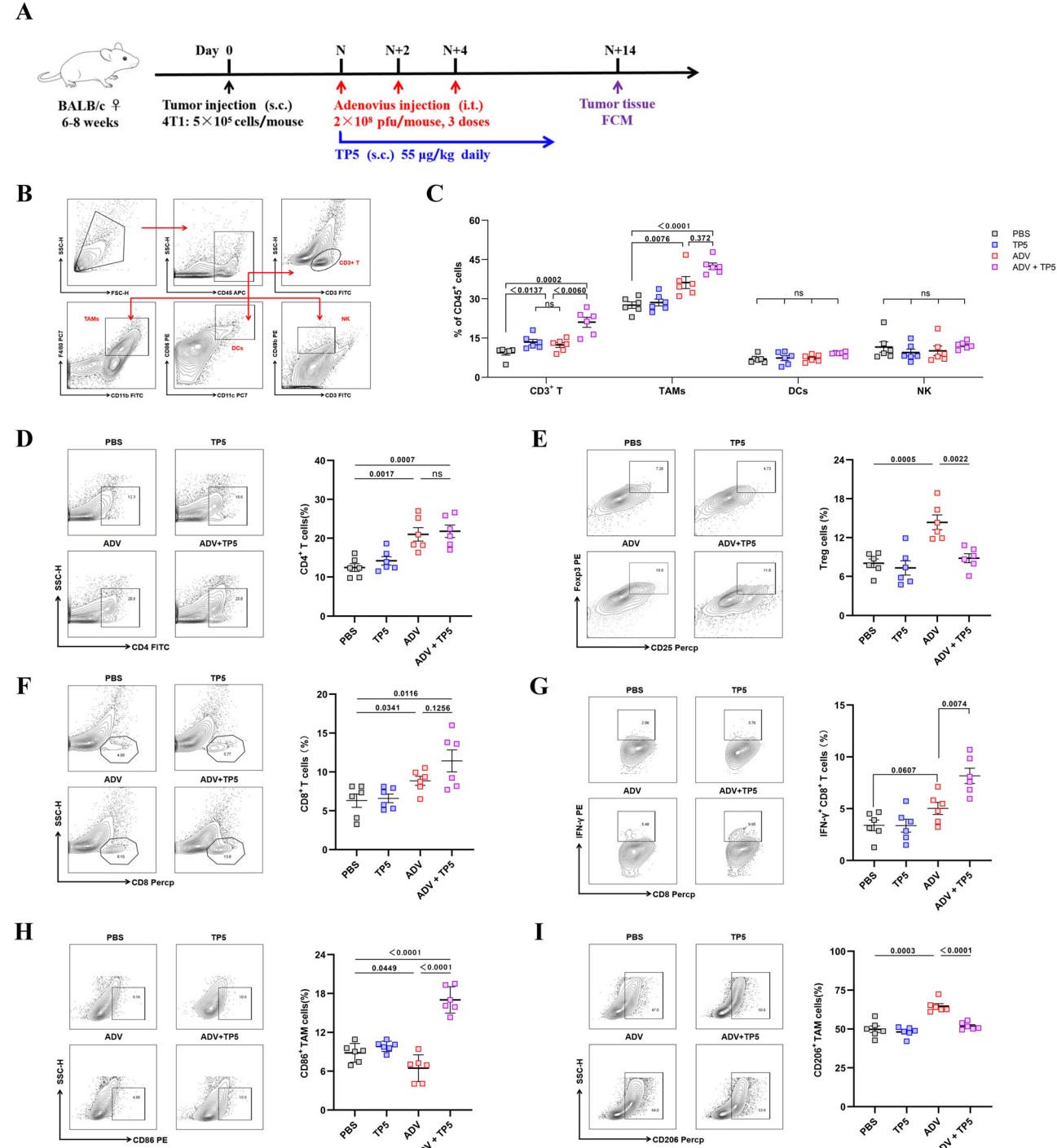

**Fig. 2 | During oncolytic virotherapy, TP5 further increases the number of proinflammatory TAMs and cytotoxic T cells in the tumors of 4T1 tumor-bearing mice. A** Schematic of the treatment schedules and dosing. **B** Flow cytometry gating strategy for the identification of immune cells within the tumors of 4T1 tumor-bearing mice. **C–I** 4T1 tumor-bearing mice were administered PBS, TP5, ADV or ADV combined with TP5. Tumor tissues from mice 14 days after different treatments were administered, and immune cells from the tumor tissues were assessed via flow cytometry. **I** The percentages of CD3+ T cells, CD11b+F4/80+ macrophages, DCs and NK cells within the tumors of the mice were monitored. Representative plots (left) and percentages (right) of tumor-infiltrating leukocytes. **D** CD4+ T cells, **E** CD25+Foxp3+ T cells, **F** CD8+ T cells, **G** IFN-γ+CD8+ T cells, **H** "M1-like" macrophages (CD86+ TAMs), and **I** "M2-like" macrophages (CD206+ TAMs) within the tumors of the mice (n = 6 mice per group) were monitored. Data are presented as mean ± SD., and P values were calculated using one-way ANOVA with Tukey's multiple comparisons test in (**C–I**). ns not statistically significant.

resulted in superior antitumor effects in the H22 model, and tumor regression was even observed in the ADV combined with TP5 group (Fig. 1H–J). Together, these results indicate that TP5 enhances the ability of ADV to suppress the progression of solid tumours, and that their anti-tumour efficacy is generalizable across multiple tumour models.

**The combination of ADV and TP5 increased the number of proinflammatory macrophages and cytotoxic T cells in the TME**
Since ADV and TP5 have been shown to regulate the biological functions of immune cells[31,36], we further investigated whether the antineoplastic effects of ADV and TP5 on solid tumors could be attributed to TME

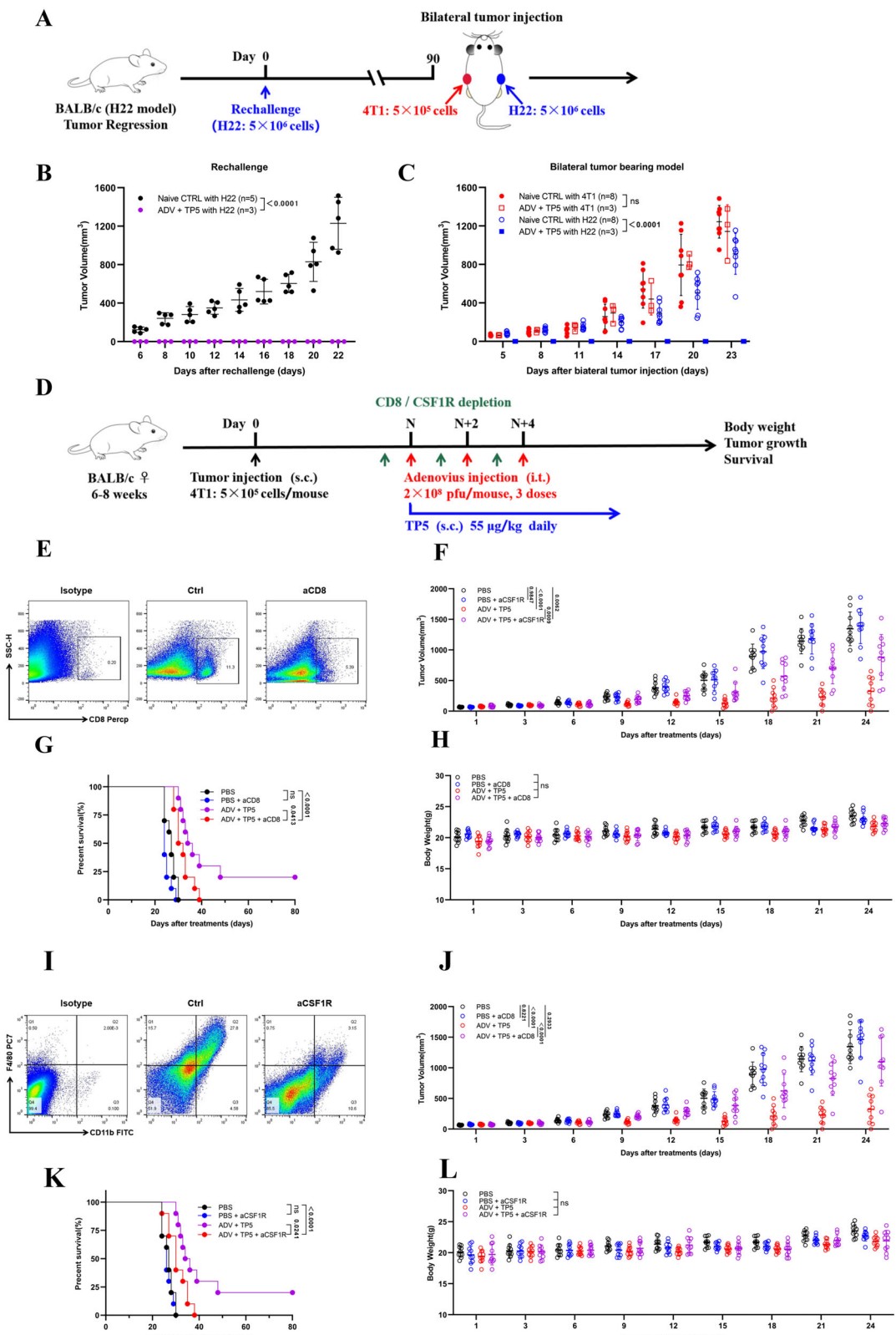

reprogramming. Thus, a 4T1 subcutaneous tumor model was constructed again, and tumor tissues were harvested 14 days after treatment (Fig. 2A). By analyzing the infiltrating immune cells in the TME of each group via flow cytometry, we found that the percentages of CD3$^+$ T cells and CD11b$^+$F4/80$^+$ macrophages were different between the treatment groups and the PBS group, but the percentages of DCs and NK cells were not different among the treatment groups in 4T1 tumor-bearing mice (Figs. 2B, C).

Next, we further explored the effects of ADV and TP5 on tumor-infiltrating T cells and TAMs. ADV alone increased the number of CD4$^+$ T and CD8$^+$ T cells, but it also led to the expansion of CD25$^+$Foxp3$^+$ regulatory T cells (Tregs) in the 4T1 model (Fig. 2D–F). Importantly, compared with the other treatments, ADV combined with TP5 had the greatest effect on increasing immune cell infiltration (Fig. 2B–F). Furthermore, we observed higher percentages of tumor-infiltrating IFN-γ-producing CD8$^+$

**Fig. 3 | Macrophages and CD8⁺ T cells mediate the antitumor activity of combination therapy. A–C** H22 model: First rechallenge model and bilateral tumor-bearing model: The cured mice that were administered immunotherapies were injected subcutaneously with $5 \times 10^6$ H22 cells. The cured mice with immunological memory were again rechallenged with $5 \times 10^6$ H22 cells, and the opposing flank was injected subcutaneously with $5 \times 10^5$ 4T1 cells on day 80 after the first rechallenge; naive mice were used as a control. **A** Experimental schematic. **B** Tumor growth in the first rechallenged model mice ($n = 5$ mice for the naive group; $n = 3$ mice for the cured/immune memory group). **C** Tumor growth in bilateral tumor-bearing model mice ($n = 8$ mice for the naive group; $n = 3$ mice for the cured/immune memory group). **D–L** BALB/c mice were subcutaneously injected with 4T1 tumor cells. When the tumor volume reached ~50–100 mm³, the mice were randomly divided into different groups ($n = 10$ per group) and treated with different immunotherapies, with or without anti-CSF1R or anti-CD8a. TP5 was injected subcutaneously once a day after the first dose of ADV was administered. Anti-CSF1R (20 mg/kg) or anti-CD8a (10 mg/kg) antibodies were administered on day 8 and then every 2 days for a total of three times. **D** Experimental timeline for the 4T1 model. **E, I** Tumor tissue samples were collected from the mice on day 15, and flow cytometry confirmed the depletion effects of the anti-CD8a (**E**) and anti-CSF1R (**I**) antibodies. **F, J** Tumor growth data are presented as the mean ± SD. **G, K** Kaplan–Meier survival curves for all groups. **H, L** Body weight data are presented as the mean ± SD. Data are presented as mean ± SD, and $P$ values were calculated using the two-way ANOVA with Geisser-Greenhouse correction (**B, C, F, H, J, L**), log-rank test in (**G, K**). ns no significant difference.

T cells in the ADV and TP5 combination treatment groups than in the other treatment groups (Fig. 2G). With respect to the effects of ADV and TP5 on macrophages, we found that ADV could lead to downregulation of CD86 and upregulation of CD206 in TAMs from ADV-treated mice compared with those from PBS- and TP5-treated mice, but ADV-induced immuno-suppressive macrophages could be reprogrammed by subcutaneous injection of TP5 (Fig. 2H, I). Notably, we found that, compared with PBS alone, subcutaneous TP5 alone did not affect the number of Treg cells, CD86⁺ TAMs, or CD206⁺ TAMs in the TME, suggesting that the effects of ADV and TP5 combination treatment are different from those of ADV or TP5 treatment alone (Fig. 2E, H, I).

In summary, these data clearly demonstrated the contribution of TP5 during combination therapy and indicated that ADV combined with TP5 inhibits the growth of solid tumors, likely by increasing the number of proinflammatory TAMs and cytotoxic T cells in the TME.

## The cured mice in the ADV and TP5 combination treatment group exhibited tumor-specific and long-term immune memory, and the antitumor effect was mediated by macrophages and CD8⁺ T cells

Tumor regression was observed in the ADV and TP5 combination treatment group on day 35 after tumor injection. To elucidate whether tumor regression mediated by this combination treatment involves immunological memory, the cured mice were again injected with tumor cells (Fig. 3A). No tumor burden was observed in the ADV and TP5 combination treatment group, suggesting that ADV combined with TP5 induced long-term immunological memory (Fig. 3B). Next, H22 and 4T1 cells were inoculated subcutaneously into mice with immunological memory to verify whether the antitumor immunity induced by ADV and TP5 combination treatment was tumor specific. We found that 4T1 tumors rapidly developed in all the mice, whereas the group treated with ADV and TP5 with immunological memory did not develop H22 tumors (Fig. 3C).

Considering the changes in the macrophage phenotype and CD8⁺ T-cell recruitment induced by ADV combined with TP5, to confirm their critical roles during treatment, we depleted macrophages or CD8⁺ T cells in 4T1-bearing mice with anti-CSF1R or anti-CD8a antibodies, respectively (Fig. 3D). Flow cytometry revealed that, compared with those in the tumor tissues of control mice, macrophages and CD8⁺ T cells were rarely found in the tumor tissues of mice after antibody injection (Fig. 3E, I). By observing tumor growth and survival in mice, we found that the depletion of macrophages and CD8⁺ T cells impaired the efficacy of the ADV and TP5 combination treatment (Fig. 4F–H, J–L).

In summary, these findings reveal that the antitumor effect induced by the combination treatment is tumor-specific and confirm that the combination of ADV and TP5 exerts its tumor-suppressing effects by regulating macrophages and CD8⁺ T cells.

## Construction and characterization of an recombinant adenovirus expressing TP5

To maximize the antitumor efficacy of ADV combined with TP5 and reduce the burden of daily administration, we genetically modified ADV and constructed a recombinant adenovirus expressing TP5 (RKDVY), which was termed ADV-TP5 (Fig. 4A).

First, we detected the secretion of TP5 in the supernatants of 4T1 cells infected with ADV-TP5 via western blotting (Fig. 4B). Next, we evaluated the oncolytic ability and replication capacity of ADV-TP5 in tumor cells. ADV or ADV-TP5 induced similar and dose (MOI)-dependent oncolytic activities in 4T1, CT26, and H22 cells, and the virus replication capacity was confirmed by the expression of E1A mRNA at different times post infection in multiple types of tumor cells. The oncolytic ability and replication capacity of the two OVs were similar (Fig. 4C–H). In addition, we measured the expression of EGFP in different cells via flow cytometry. The related results revealed that ADV and ADV-TP5 had similar replication capacities in both mouse and human cells and that ADV and ADV-TP5 specifically replicated in tumor cells, resolving important safety concerns (Fig. 4I). Similarly, when tumour cells were infected with ADV-TP5 at increasing MOIs, the proportion of GFP⁺ cells increased in an MOI-dependent manner, indicating that ADV-TP5 efficiently infected and replicated in tumour cells (Fig. S1A–C). Next, 4T1 cells were collected 48 h post infection and stained with Annexin-V/PI. Compared with the control, both ADV and ADV-TP5 effectively induced early and late apoptosis (and/or necrosis) in tumor cells (Fig. 4J). We further evaluated the in vivo behavior of ADV-TP5 (Fig. S1D–G). Consistent with the in vitro results, ADV-TP5 effectively infected and replicated within tumour tissues while expressing TP5.

Collectively, these results reveal that TP5 gene insertion does not impair the replication and oncolytic capabilities of the virus and that ADV-TP5 is a functional oncolytic adenovirus.

## ADV-TP5 suppresses tumor growth in solid tumor models through the regulation of CD8⁺ T cells and macrophages

To investigate the antitumor activity of ADV-TP5 in vivo, we treated tumor-bearing mice with different treatments (Fig. 5A). We found that ADV-TP5 was capable of significantly suppressing tumor growth and prolonging survival in 4T1 tumor-bearing mice and that its antitumor efficacy was no lower than that of ADV combined with TP5 (Fig. 5B, C). Similarly, ADV-TP5 also had superior antitumor effects on the hepatocellular carcinoma (HCC) model (Fig. 5E, F). Additionally, the different treatments did not have significant toxic effects on the mice (Fig. 5D, G). To determine whether the antitumour effect of ADV-TP5 depends on viral replication, we used inactivated virus to exclude the contribution of viral capsid proteins. Inactivated ADV-TP5 failed to exert antitumour activity (Fig. S2A–C) and likewise did not induce an effective antitumour immune response (Fig. S2D–G).

Next, we depleted CD8⁺ T cells or macrophages in 4T1 tumor-bearing mice via anti-CD8a or anti-CSF1R antibodies, respectively (Fig. 5H). As expected, the antitumor efficacy of ADV-TP5 was sharply diminished by depletion of CD8⁺ T cells and/or macrophages with no drug toxicity (Fig. 5I–K). To further delineate the mechanism underlying the antitumour activity of ADV combined with TP5, we concomitantly depleted both macrophages and CD8⁺ T cells to determine whether this would completely abrogate the therapeutic effect. Interestingly, simultaneous depletion of both CD8⁺ T cells and macrophages did not result in a significantly greater

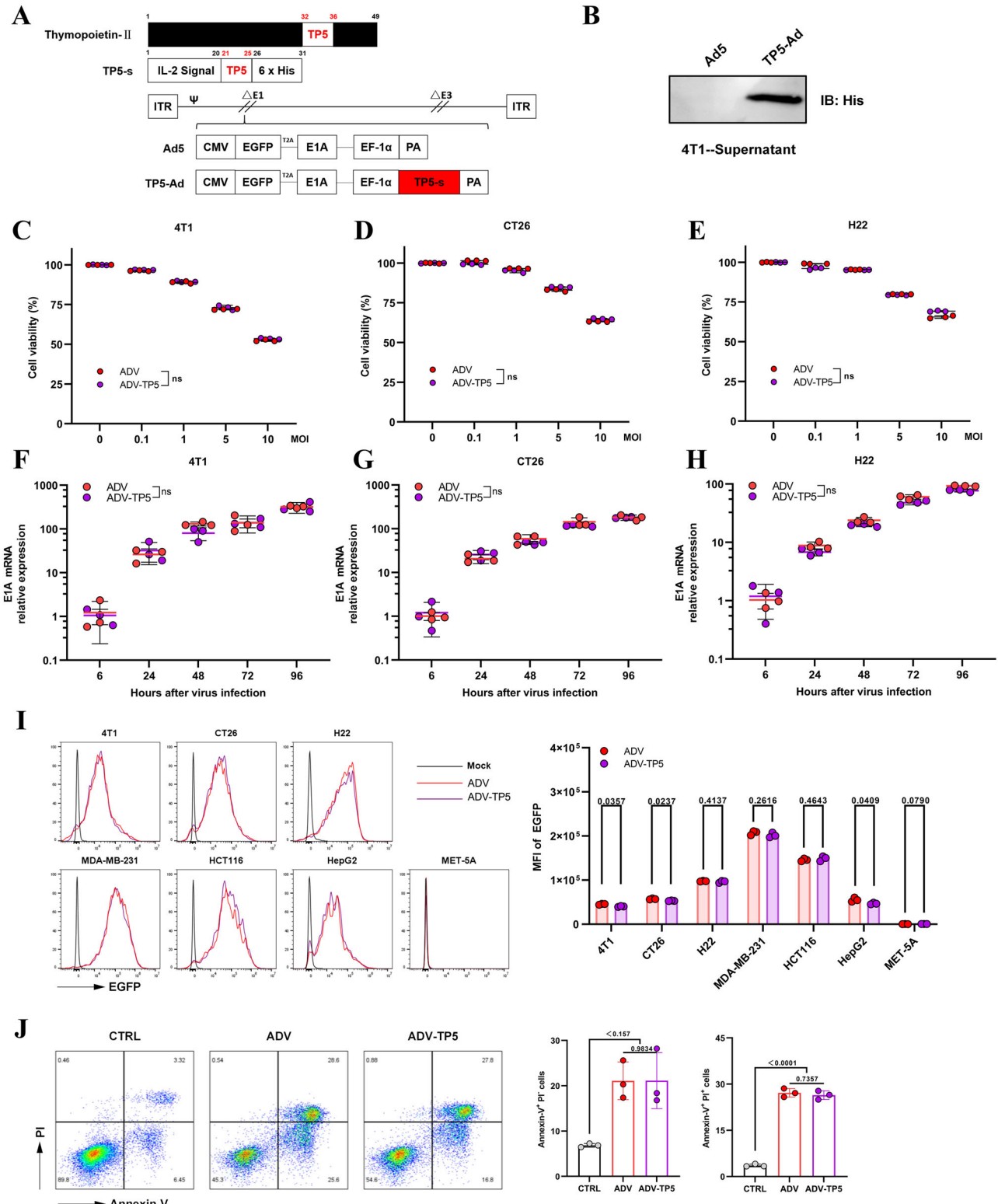

**Fig. 4 | Construction and characterization of the recombinant adenovirus ADV-TP5. A** Construction of the recombinant adenovirus ADV-TP5. **B** 4T1 cells were infected with ADV or ADV-TP5 (MOI = 2) for 48 hours, and western blotting confirmed the secretion of TP5 in the supernatant. **C–E** 4T1 cells (**C**), CT26 cells (**D**) and H22 cells (**E**) were infected with ADV or ADV-TP5 at different MOIs, and cell viability was measured 48 h later via a CCK-8 assay (*n* = 3 per group). **F–H** 4T1 cells (**F**), CT26 cells (**G**) and H22 cells (**H**) were infected with different titers of ADV or ADV-TP5 and harvested at different time points post infection. The viral copy number was determined via qPCR. The fold changes were calculated by dividing the copy number at 6 h (*n* = 3 per group). **I** 4T1 cells, CT26 cells, H22 cells, MDA-MB- 231 cells, HCT116 cells, HepG2 cells and MET-5A cells were infected with ADV or ADV-TP5 at an MOI of 1. The expression of EGFP in the cells was measured by flow cytometry (*n* = 3 per group). **J** 4T1 cells were infected with ADV or ADV-TP5 (MOI = 2) for 48 h, the early apoptotic cells were confirmed as Annexin-V$^+$/PI$^-$ cells, and the late apoptotic (and/or necrotic) cells were confirmed as Annexin-V$^+$PI$^+$ cells (*n* = 3 per group). Data are presented as mean ± SD, and *P* values were calculated using the two-way ANOVA with Geisser-Greenhouse correction (**C–H**), two-tailed Student's *t* test in (**I**) or one-way ANOVA with Tukey's multiple comparisons test in (**J**). ns not statistically significant.

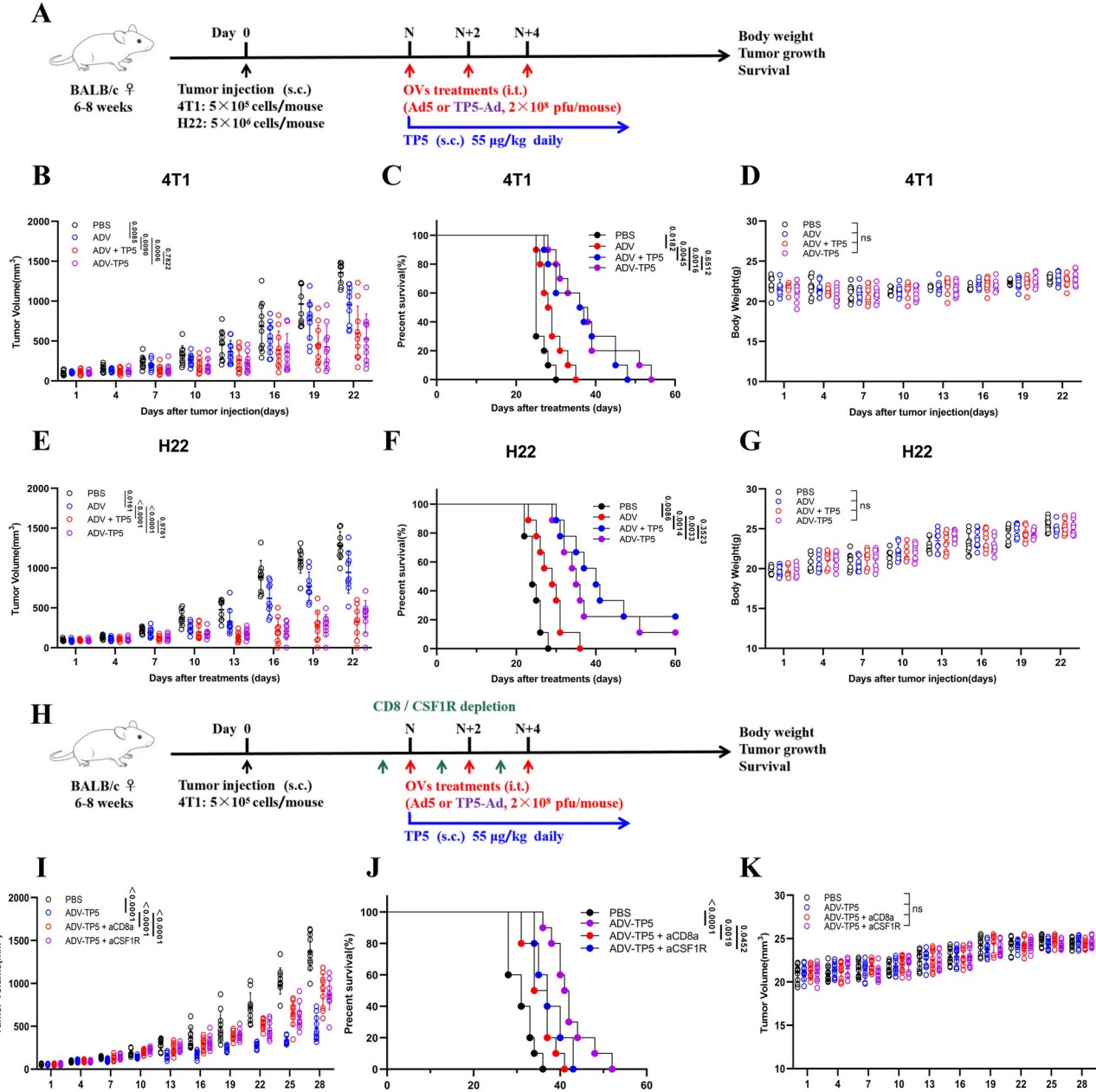

**Fig. 5 | ADV-TP5 suppresses tumor growth in solid tumor models through the regulation of CD8⁺ T cells and macrophages.** **A** The therapeutic schedule for different treatments in tumor-bearing mice. BALB/c mice were subcutaneously injected with tumor cells. When the tumor volume was ~50–100 mm³, the mice were randomly divided into different groups and treated with PBS, ADV or ADV-TP5 three times every 2 days. TP5 was injected subcutaneously into the peritumoral site once daily, starting with the first dose of ADV or PBS. **B–D** 4T1 model (*n* = 10 mice per group), **B** tumor growth. **C** Kaplan–Meier survival curves for all groups. **D** Body weight. **E–G** H22 model (*n* = 10 mice per group), **E** tumor growth. **F** Kaplan–Meier survival curves for all groups. **G** Body weight. **H–K** BALB/c mice were

subcutaneously implanted with 4T1 tumor cells. When the tumor volume reached ~50–100 mm³, the mice were randomly divided into different groups (*n* = 10 per group) and treated with different immunotherapies, with or without anti-CSF1R or anti-CD8a. Anti-CSF1R (20 mg/kg) or anti-CD8a (10 mg/kg) antibodies were administered on day 8 and then every 2 days for a total of three times. **H** Experimental timeline for the 4T1 model. **I** Tumor growth. **J** Kaplan–Meier survival curves for all groups. **K** Body weight data are presented as the mean ± SD. Data are presented as mean ± SD, and *P* values were calculated using the two-way ANOVA with Geisser-Greenhouse correction (**B, D, E, G, I, K**), log-rank test in (**C, F, J**). ns no significant difference.

impairment of therapeutic efficacy compared to individual depletions (Fig. S3A–C). Flow cytometric analysis showed that CD8⁺ T cell depletion reduced the infiltration of M1 macrophages, whereas macrophage depletion likewise diminished the infiltration of cytotoxic CD8⁺ T cells (Fig. S3D–F). This suggests that both cell types operate in a complementary rather than completely independent manner in mediating the antitumor effects, and that eliminating either population is sufficient to substantially disrupt the therapeutic mechanism. These data reveal that ADV-TP5 is a promising

immunotherapy agent and that CD8⁺ T cells and macrophages are indispensable for ADV-TP5-induced antitumor immunity.

## ADV-TP5 reprograms the TME toward a more beneficial state for antitumor immunity

To evaluate the immunomodulatory capacity of ADV-TP5 in vivo, tumor-infiltrating immune cells in each treatment group were analyzed on day 14 after different treatments (Fig. 6A). We observed that the percentages of 4T1

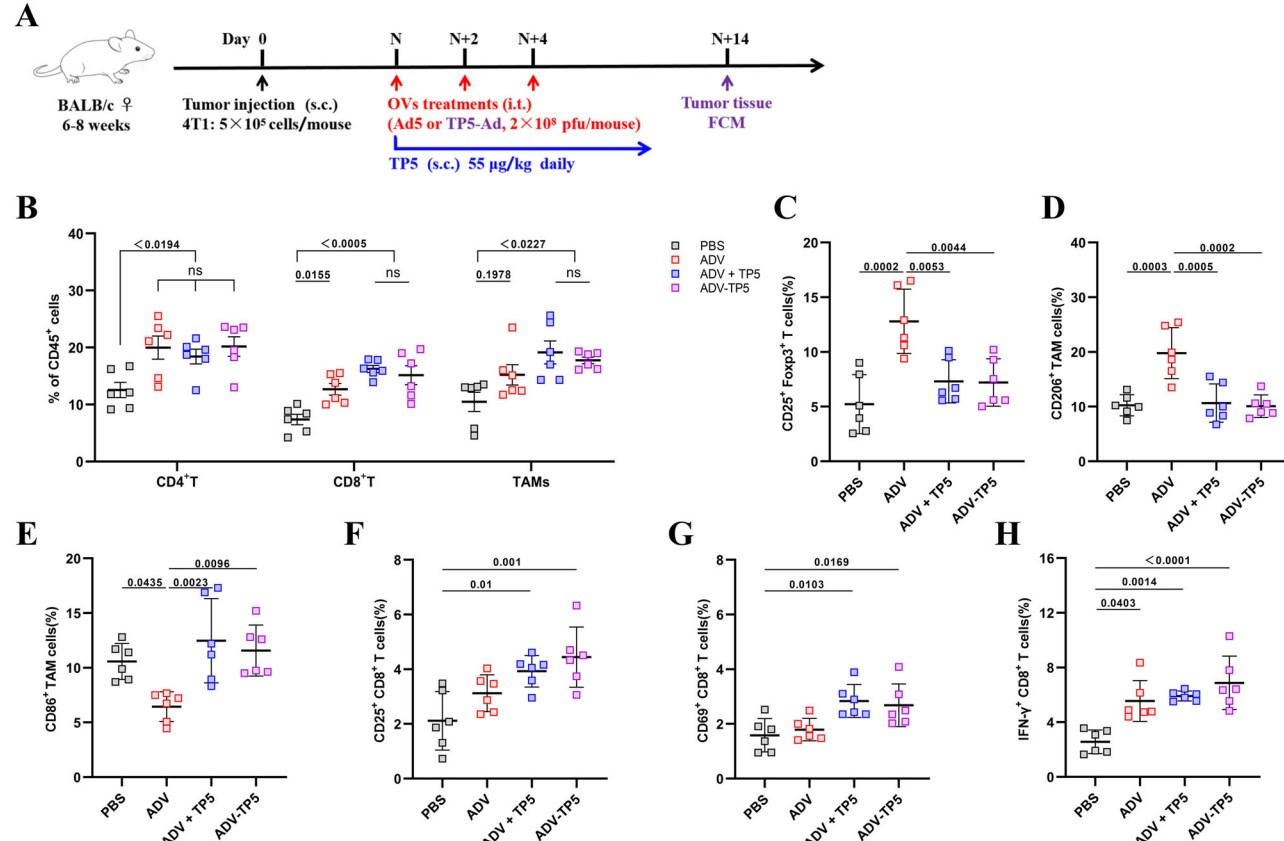

**Fig. 6 | ADV-TP5 reprograms the TME toward a more beneficial state for anti-tumor immunity. A** Schematic of the treatment schedules and dosing. **B–H** 4T1-bearing mice were subjected to different treatments. Tumor tissues from mice 14 days after different treatments were generated, and immune cells from the tumor tissues were assessed via flow cytometry (*n* = 6 mice per group). **B** Percentages of CD4$^+$ T cells, CD8$^+$ T cells and CD11b$^+$F4/80$^+$ macrophages within the tumors of the mice were monitored. The percentages of tumor-infiltrating **C** Tregs, **D** "M2-like" macrophages (CD206$^+$ TAMs), **E** "M1-like" macrophages (CD86$^+$ TAMs), **F** CD25$^+$CD8$^+$ T cells, **G** CD69$^+$CD8$^+$ T cells and **H** IFN-γ$^+$CD8$^+$ T cells within the tumors of the mice (*n* = 6 mice per group) were monitored. Data are presented as mean ± SD, and *P* values were calculated using one-way ANOVA with Tukey's multiple comparisons test in (**B–H**). ns not statistically significant.

tumor-infiltrating immune cells, including CD3$^+$ T cells, CD8$^+$ T cells and macrophages, were significantly greater in the ADV-TP5 group and the ADV and TP5 combination treatment group than in the other groups. Moreover, the ADV-TP5 group presented the most abundant tumor immune cell infiltration among all of the groups of mice (Fig. 6B). Compared with our previous observations, we found that the percentage of CD86$^+$ TAMs was significantly decreased, whereas the percentages of CD206$^+$ TAMs and Tregs were markedly increased in 4T1 tumors from the ADV group compared with those from the PBS group, and the ADV-induced TAM phenotype could be reversed by intervention with TP5. In addition, the highest percentage of CD86$^+$ TAMs and the lowest percentages of CD206$^+$ TAMs and Tregs were observed in the ADV-TP5 group (Fig. 6C–E).

To further explore whether ADV-TP5 could enhance the antitumor activity of T cells, we analyzed the subtypes of CD8$^+$ T cells in the 4T1 model. As expected, we found that the expression of CD25, CD69, and IFN-γ was the highest in the CD8$^+$ T cells of the ADV-TP5 group (Fig. 6F–H). In summary, the recombinant oncolytic adenovirus ADV-TP5 can effectively orchestrate the reprogramming of tumor-associated macrophages and activate CD8$^+$ T cells to exert superior antitumor activity in vivo.

### ADV-TP5 has strong therapeutic effects in humanized mouse models, and the enhancing effect of TP5 on oncolytic virotherapy efficacy is universal regardless of the OV used

To evaluate the clinical translational potential of ADV-TP5, we tested the antitumor activity of ADV-TP5 in humanized mouse models. Tumor-bearing NCG mice (nonobese diabetic-SCID IL-2 receptor gamma null

mice) were intravenously injected with human peripheral blood mononuclear cells on day 1 so that the humanized mouse tumor model was successfully constructed. Consistent with previous results, tumor growth was inhibited in the mice that received ADV-TP5 (Fig. 7A). Moreover, ADV-TP5 treatment led to significantly prolonged survival in both the TNBC model and the HCC model (Fig. 7B). More importantly, ADV-TP5 treatment also inhibited tumor progression in the PDX TNBC model (Fig. 7C–E). These results indicated that ADV-TP5 was sufficient to activate human immune cells and induce antitumor responses across various mouse models.

Finally, we aimed to explore whether TP5 combined with other types of OVs could also produce antitumor immune responses that effectively activate immune cells. To this end, herpes simplex virus (HSV) and vaccinia virus (VV) were used to treat 4T1 tumor-bearing mice. We found that TP5 enhanced the antitumor efficacy of these two OVs, significantly inhibited tumor growth and prolonged survival in mice (Fig. 7F, G). Similarly, flow cytometric analysis showed that both HSV and VV combined with TP5 increased the infiltration of M1-like macrophages and cytotoxic CD8$^+$ T cells (Fig. S4). Taken together, TP5 has the ability to enhance the efficacy of oncolytic virotherapy with various OVs to achieve superior antitumor effects, and ADV-TP5 is a potential tumor therapy drug.

### Discussion

Here, we identified a suitable and economical agent (TP5) for increasing the antitumor efficacy of OVs. In a variety of solid tumors, ADV combined with TP5 significantly increased immune cell infiltration into the TME. Compared with monotherapy, combination therapy had superior antitumor

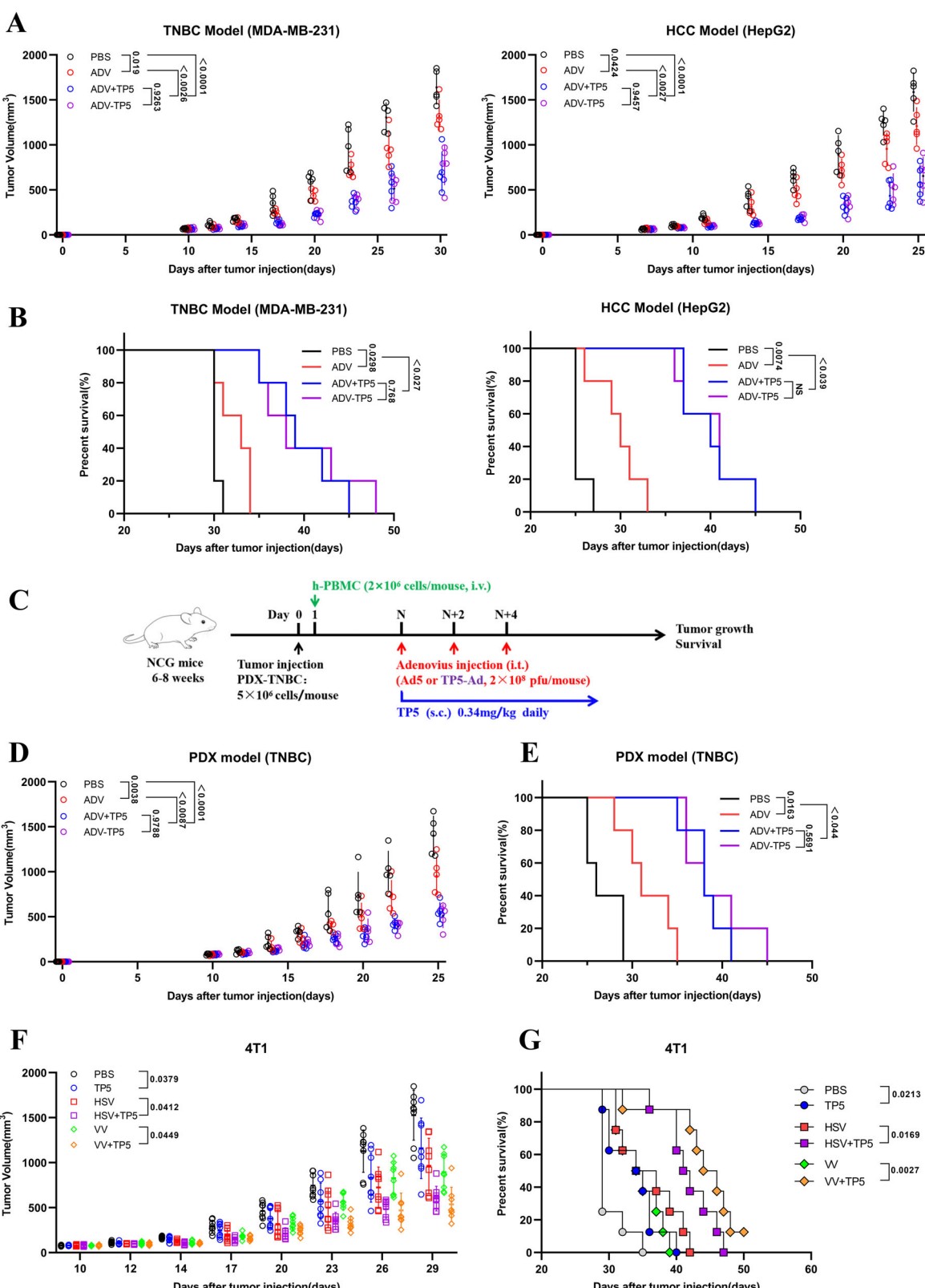

effects, and some tumor-bearing mice were cured after treatment with ADV combined with TP5. On the basis of these results, we constructed an oncolytic adenovirus, ADV-TP5, via genetic modification[36]. As expected, ADV-TP5 treatment significantly inhibited tumor growth and prolonged survival in both murine and humanized tumor models because of the ability of ADV-TP5 to effectively reprogram the TME into a state that is more

beneficial to antitumor immunity. More importantly, TP5 can combine with multiple OVs to produce more effective antitumor immune responses, suggesting the clinical potential of TP5 in oncolytic virotherapy (Fig. 8).

Since T-VEC was approved by the US Food and Drug Administration (FDA) for the treatment of advanced metastatic melanoma, oncolytic virotherapy has received much attention in the field of tumor

**Fig. 7 | ADV-TP5 has strong therapeutic effects in humanized mouse models, and TP5 has the ability to enhance the efficacy of HSV and VV oncolytic virotherapy.** **A**, **B** Tumor-bearing NCG mice were intravenously injected with human peripheral blood mononuclear cells on day 1, after which the mice were administered different immunotherapies ($n = 5$ mice per group). **A** Tumor growth data are presented as the mean ± SD in both TNBC model (left) and HCC model (right) mice, and *P* values were determined by one-way ANOVA. **B** Kaplan–Meier survival curves for all groups of both TNBC model (left) and HCC model (right) mice. **C–E** PDX tumor cells were injected into the fourth mammary fat pads of NCG mice, and these tumor-bearing mice were intravenously injected with human peripheral blood mononuclear cells on day 1, followed by different immunotherapies starting on the day when the tumor volume was ~50–100 mm³ ($n = 5$ mice per group). **C** Schematic of the treatment schedules and dosing. **D** Tumor growth data are presented as the mean ± SD, and *P* values were determined via one-way ANOVA. **E** Kaplan–Meier survival curves for all groups. **F**, **G** BALB/c mice were subcutaneously implanted with 4T1 tumor cells. When the tumor volume was ~50–100 mm³, the mice were randomly divided into different groups and treated with PBS, HSV or VV three times every 2 days. TP5 was injected subcutaneously into the peritumoral site once daily, starting with the first dose of OVs or PBS ($n = 8$ mice per group). **F** Tumor growth data are presented as the mean ± SD, and *P* values were determined via one-way ANOVA. **G** Kaplan–Meier survival curves for all groups. Data are presented as mean ± SD, and *P* values were calculated using the two-way ANOVA with Geisser-Greenhouse correction (**A**, **D**, **F**), log-rank test in (**B**, **E**, **G**). ns no significant difference.

**Fig. 8 | A schematic diagram showing the mechanism by which TP5 powers oncolytic virotherapy by orchestrating TAMs and CD8⁺ T cell.** (Upper panel) Schematic depiction of the tumor microenvironment (TME) composition, including tumor cells, M1 and M2 tumor-associated macrophages (TAMs), CD4⁺ and CD8⁺ T cells, regulatory T cells (Treg) and cytokines. (Lower left panel) Monotherapy with ADV: Treatment with oncolytic adenovirus (ADV) alone induces only partial immunosuppression reversal within the TME, resulting in limited antitumor efficacy. (Lower right panel) Combination therapy with TP5: Combination of oncolytic adenovirus with TP5 (Ad5+TP5 or ADV-TP5) potentiates virotherapy. TP5 orchestrates TAM polarization (shifting from M2 to M1 phenotype) and enhances CD8⁺ T cell activity, thereby remodeling the TME and augmenting antitumor immune responses. OV oncolytic virus, ADV, adenovirus, VV vaccinia virus, HSV herpes simplex virus, TP5 thymopentin, TAM tumor-associated macrophage, Treg regulatory T cell.

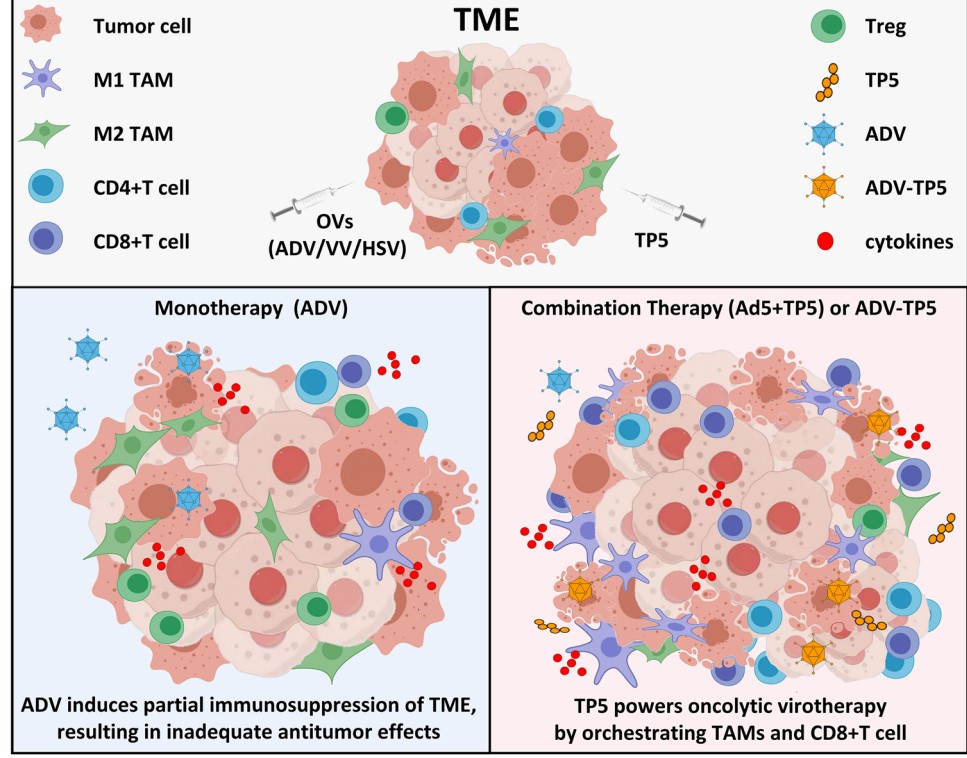

immunotherapy[37,38]. OVs can not only specifically infect and lyse tumor cells but also initiate immunogenic cell death (ICD), thus releasing pathogen-associated molecular patterns (PAMPs), damage-associated molecular patterns (DAMPs), and tumor-associated antigens (TAAs) to reprogram the TME and activate antitumor immune responses[39–41]. The findings of our study are consistent with those of previous studies demonstrating that ADV treatment could effectively transform tumors from "cold" to "hot"[23,42–44]. However, the ability of ADV to reprogram the TME does not seem to be uniformly favorable for antitumor immunity, especially since it does not regulate tumor-infiltrating Tregs or TAMs. Anti-inflammatory M2-like tumor-associated macrophages (M2-TAMs), Tregs, and bone marrow-derived suppressor cells (MDSCs) are major immunosuppressive players in the TME[45–48]. These data may explain the unsatisfactory antitumor efficacy of ADV alone.

Our findings revealed that TP5 could repolarize TAMs during ADV treatment. Specifically, both ADV combined with TP5 and ADV-TP5 decreased the expression of CD206 and increased the expression of CD86 in TAMs. With respect to CD8⁺ T cells, both ADV combined with TP5 and ADV-TP5 promoted the expression of activation and cytotoxic markers (CD25, CD69 and IFN-γ). These data demonstrate that ADV and TP5 may stimulate cytotoxic CD8⁺ T cells either directly or indirectly via the plasticity of TAMs.

In line with these findings, TIL profiling further supports that ADV + TP5 reprograms the TME in our xenograft models. Compared with either single agent, the combination increased the frequency and number of activated, IFN-γ⁺ CD8⁺ T cells and enriched effector-like T cell subsets, while reducing FOXP3⁺ regulatory T cells and thereby raising the CD8⁺ T cell/Treg ratio within tumors. Together, these changes in the TIL landscape indicate a shift from an immunosuppressive, Treg-rich microenvironment to an effector-dominant, pro-inflammatory state, which is consistent with the enhanced antitumor efficacy of Ad5 + TP5 and ADV-TP5.

Of note, the overall antitumor efficacy of ADV-TP5 was comparable to that achieved with the regimen of ADV plus exogenous TP5 in our models. Regarding this result, we believe that both strategies ultimately provide the same key effector component in the tumor microenvironment: a replicable adenovirus type 5 (ADV) backbone and TP5 at pharmacologically effective levels. Our in vitro data indicate that inserting TP5 does not impair viral replication or oncolytic activity, suggesting that the backbone viruses in both groups are functionally equivalent. Under the dosing regimen used in this study, both ADV combined with TP5 and ADV-TP5 likely deliver sufficient

effector TP5, resulting in similar inhibition of tumor growth and survival. Therefore, we consider the main advantage of ADV-TP5 to be the integration of TP5 delivery into a single oncolytic virus formulation, which ensures sustained local release while avoiding repeated peptide injections, potentially improving treatment convenience and translational feasibility.

There are several directions to explore in the future. Our study focused on the effects of ADV combined with TP5, whereas the individual effects of ADV and TP5 have not been studied in detail. For example, we do not explore why ADV treatment leads to an increase in the numbers of Treg cells and M2 TAMs, and the specific molecular mechanisms by which TP5 regulates macrophages are unclear. In addition, understanding why TP5 can enhance the efficacy of different OVs may provide further guidance for the design of oncolytic therapies.

In conclusion, oncolytic virotherapy, represented by ADV, can recruit many immune cells to infiltrate the tumor site, and TP5 can effectively regulate immune cells to reprogram the TME into a state more beneficial for the antitumor immune response, thereby inhibiting tumor progression. This work provides a foundation for accelerating the development of virotherapies and their combination regimens in preclinical models and patients. Furthermore, this study has clinical translational prospects: (1) TP5 has been clinically approved in many countries worldwide as an immunomodulator, (2) The potentiating effect of TP5 in oncolytic virotherapy appears to be broadly generalizable, and (3) the antitumor activity of the combination therapy and ADV-TP5 has been found in multiple humanized tumor models.

## Data availability

The authors declare that the main data supporting the findings of this study are available within the article and its Supplementary Information files. Extra data are available from the corresponding author upon request.

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

## Acknowledgements

The research was supported by the Key R&D Program of Shandong Province, China (2025CXPT176); the Shandong Provincial Natural Science Foundation (ZR2025MS1306); the National Natural Science Foundation of China (81972888, 82272819); the Research Project of Jinan Microecological Biomedicine Shandong Laboratory (JNL-2025008B, JNL-2025009B, JNL-2025011B, JNL-2025010B, JNL-2025012B, and JNL-2023017D); the Shandong Provincial Laboratory Project (SYS202202); and the Primary Research & Development Plan of Jiangsu Province (BE2022840). We thank the Research Center for Basic Medical Science of Nanjing University Medical School for their technical support.

## Author contributions

Conceptualization: J.W., C.J. and L.K.; methodology: L.K., K.L., Y.L., J.W., and C.J.; investigation: L.K., K.L., H.C., Y.L., P.W., J.Q., Q.X., D.Z., W.L., F.Z. and J.W.; visualization: K.L. and L.K.; funding acquisition: J.W., C.J. and X.G.; project administration: J.W. and C.J.; supervision: J.W. and C.J.; writing—original draft: K.L. and J.W.; writing—review and editing: J.W., C.J. and L.K.

## Competing interests

K.L., J.W., C.J., L.K., H.C., X.G. and Q.X. are named as inventors on the China patent application (no. 2024110168233), which is related to this work. Jinan Microecological Biomedicine Shandong Laboratory and Nanjing University are the applicants of this patent application. X.G. and Q.X. are employees of Jinan Microecological Biomedicine Shandong Laboratory. J.W. and C.J. are employees of Nanjing University. The remaining authors declare no competing interests.
