## [Transparent Peer Review file · Communications Medicine]

Thymopentin enhances adenoviral oncolytic therapy by regulating macrophages and CD8⁺ T cells

Corresponding Author: Professor Junhua Wu

Version 0:

Reviewer comments:

Reviewer #1

(Remarks to the Author)

The authors have submitted a manuscript discussing a combination treatment involving thymopentin (TP5) and oncolytic adenovirus targeting tumor cells. They propose that TP5 modulates immune cell populations, including macrophages and CD8⁺ T cells, thereby improving survival. The manuscript is clearly written, and the findings are significant for the field of oncolytic virotherapy. While there are a few concerns, I do not believe they preclude publication in *Communication Medicine*. However, the following minor revisions should be addressed:

1. Please consult with a statistician regarding the statistical analyses. Several aspects of the statistical methodology are unclear. It would be helpful to describe the statistical methods used for each figure in more detail.
2. I did not see a Dead/live gating strategy described in Figure 2. Was live/dead staining included in the procedure? Additionally, which control group was used—Isotype control or FMO?
3. Could the authors present CD4⁺ and CD8⁺ T cell data in a single graph for easier comparison in Figure 2D and F?
4. The results from subcutaneous and intratumoral injections, including tumor volume and survival, appear quite similar. Could the authors explain how these findings support their conclusion? Also, is there a method to quantify TP5 expression after TP5-Ad injection?
5. The authors use the term 'synergistically.' Please clarify how synergy was defined or calculated in this context.
6. The manuscript refers to HSV, VV, and Ad5 showing similar effects. Please provide more detail about these viruses and clarify why similar responses were observed with three different oncolytic viruses.
7. In Figure 2, IFN- γ ⁺ CD8⁺ T cells in the PBS and Ad5 groups are not significantly different, but in Figure 6, a significant difference is reported. Please explain this discrepancy and re-check your raw flow cytometry data.

Minor Error:

In Figure 5H, it appears that groups with neither Ad5 nor TP5 s.c. injection are shown. Please remove this data from the figure to avoid confusion.

Reviewer #2

(Remarks to the Author)

Authors combine adenovirus Ad5 with an immune regulatory peptide (thymopoietin 5, TP5) to enhance the immune rejection of tumors. They clearly demonstrate that the combination increases T cell infiltration and proinflammatory macrophages. Immune responses against the tumor are demonstrated by rejection of tumor rechallenges. In a second part, a recombinant virus (TP5-Ad) is constructed expressing GFP and E1a under the CMV promoter and TP5 (with signal peptide for secretion and 6His tag to be detected) under the EF1 promoter. Note that this virus is not tumor selective (E1a is under the CMV promoter, which is not tumor-selective) and that the mere transduction (entry) of this virus will induce GFP and TP5 expression (this virus works as a regular expression vector, where the transgenes are expressed without the need of any replication). It would have been much better to have a really oncolytic (tumor-selective) adenovirus.

Major issues

Mouse cells are not permissive to human adenoviruses. Authors should quantify levels of infection (transduction or entry of virus) and replication (production of progeny virus) in the three mouse cell lines used (4T1, CT26 and H22). Infectivity or transduction can be done in vitro infecting cells at different MOIs with reporter vectors (TP5-Ad for example) and counting the amount or percentage of transduced cells. Progeny production (burst size) can be measured after infection at different MOIs, washing, and incubating for several days to measure the PFUs produced.

In the experiments with the TP5-Ad (Fig 4) 4T1 cells infected at MOI=2 produce some TP5 secreted protein (Fig4B) which implies that some (unknown) level of transduction is achieved. Still, the level of transduction (percentage of transduced cells) at this MOI and other MOIs should be evaluated for the three models. GFP intensity (Fig4) does not quantify the transduction levels and E1a expression or cytotoxicity does not demonstrate viral replication (despite authors assume it). How many cells (percentage) are GFP+ at different MOIs? How much progeny virus is produced?

In a similar way, in the in vivo experiments should be an evidence that oncolysis (viral replication) is happening in the mouse tumors. This is needed to demonstrate that the therapeutic and tumor remodeling effect is associated to tumor cell infection and oncolysis. For this, the amount of viral infection (hexon staining) obtained after the intratumoral injections should be evaluated. Without such evidence one can assume that the inflammatory and tumor remodeling effect of the virus is due to the injected capsids, without any need of tumor cell infection. A control with inactivated virus (or empty capsids) should be included in the experiments.

Minor comments

Introduction

“Thymopentin II, a natural 49-amino acid polypeptide, is isolated from the thymus^{25,26}. TP5, an active component of thymosin II, is a pentapeptide corresponding to positions 32–36 of thymosin II²⁶⁻²⁸.”

There seems to be an incoherence of thymopentin II and thymosin II names.

Results

Authors use a virus with CMV-EGFP-E1a and EF1-pA (empty cassette) as a control for their TP5-Ad virus. This virus should not be named “Ad5” .

Reviewer #3

(Remarks to the Author)

This manuscript presents an innovative approach by combining thymopentin (TP5) with adenovirus 5 (Ad5) and evaluating its therapeutic efficacy across multiple tumor models. The authors have done an exceptional job characterizing the immune mechanisms underlying the antitumor effects of this combination, particularly in challenging models such as 4T1. The experimental design is rigorous, and appropriate controls were consistently included to distinguish the effects of combination therapy from monotherapies.

While the study is comprehensive and explores multiple aspects of the therapeutic strategy, several points would benefit from further clarification and refinement. The reviewer offers the following recommendations to enhance the manuscript's clarity and scientific depth:

1. Tumor Microenvironment (TME) Reprogramming: A more detailed explanation is needed regarding how the Ad5 + TP5 combination reprograms the TME in xenograft models. Including a discussion on this point would strengthen the mechanistic insight.

2. Flow Cytometry Gating Strategy: Although gating strategies are included, clarification is needed on how specific markers (e.g., CD8 vs. CD206, CD4 vs. F4/80 vs. CD69, CD3 vs. CD11b, and FoxP3 vs. IFN- γ) were distinguished when conjugated to the same fluorochromes. This is critical for reproducibility and interpretation of the immunophenotyping data.

3. Mechanistic Depletion Studies: The manuscript shows that depletion of CD8 or CSF1R partially abrogated the combination efficacy. It is recommended to consider a dual depletion strategy (CD8 + CSF1R) to determine whether a more complete loss of therapeutic effect occurs, which could further delineate the mechanisms involved.

4. Plain Language Summary: The current plain language summary requires revision to be more accessible to non-specialist audiences. Simplifying terminology and clearly explaining the study's significance will improve its utility for broader readership.

5. Language and Formatting Edits: Minor but important language editing is needed throughout. Specific suggestions include (these are just few examples):

- Line 90 (32–36): Revise for clarity
- Line 71: Replace “good” with a more precise descriptor
- Define abbreviations at first use (e.g., HSV1 in line 79, HCC in line 130)
- Lines 101–103 in the Introduction may be better suited for the Discussion section

6. Ethical Approval Statement: The manuscript lacks mention of ethical approval for the use of human samples (e.g., PBMCs or patient-derived tumor specimens). This should be explicitly stated to ensure compliance with ethical research standards.

7. Because the work is so extensive, a graphical abstract or figure outlining the major concept would significantly help the readers.

Overall, this is a well-executed and impactful study. Addressing the points above will further improve the clarity, rigor, and translational relevance of the work.

Version 1:

Reviewer comments:

Reviewer #1

(Remarks to the Author)

The authors have addressed most of the concerns raised in the previous review. The revised manuscript demonstrates clear improvement. Overall, the revisions are satisfactory and have clarified the key points raised earlier. I have no additional major concerns.

Minor comment:

Figure 4C: Please add an x-axis label to enhance clarity.

Reviewer #2

(Remarks to the Author)

None

Reviewer #3

(Remarks to the Author)

No additional comments. Authors have sufficiently addressed all of my concerns.

Reviewers' comments:

Reviewer #1 (Remarks to the Author): The authors have submitted a manuscript discussing a combination treatment involving thymopentin (TP5) and oncolytic adenovirus targeting tumor cells. They propose that TP5 modulates immune cell populations, including macrophages and CD8⁺ T cells, thereby improving survival. The manuscript is clearly written, and the findings are significant for the field of oncolytic virotherapy. While there are a few concerns, I do not believe they preclude publication in Communication Medicine. However, the following minor revisions should be addressed:

1. Please consult with a statistician regarding the statistical analyses. Several aspects of the statistical methodology are unclear. It would be helpful to describe the statistical methods used for each figure in more detail.

Response:

We thank the reviewer for this valuable suggestion. We have consulted with a professional statistician to review and validate all statistical analyses in our study. The statistical methods for each figure have now been described in greater detail in the revised Methods section (lines 257-262). Specifically, we have clarified the use of one-way/two-way ANOVA for multiple group comparisons, Student's t-test for two-group comparisons, and the Kaplan-Meier method with log-rank test for survival analysis. All analyses were performed using GraphPad Prism software.

2. I did not see a Dead/live gating strategy described in Figure 2. Was live/dead staining included in the procedure? Additionally, which control group was used—Isotype control or FMO?

Response:

We appreciate this important observation. We have now updated Figure 2 in the revised manuscript to include a comprehensive gating strategy that explicitly shows live/dead cell discrimination using a viability dye. For all flow cytometry analyses, we used fluorescence minus one (FMO) rather than isotype controls to ensure accurate population gating and minimize background interference. This information has been

added to the Methods section (lines 192-194).

Figure 2 During oncolytic virotherapy, TP5 further increases the number of proinflammatory TAMs and cytotoxic T cells in the tumors of 4T1 tumor-bearing mice.

3. Could the authors present CD4⁺ and CD8⁺ T cell data in a single graph for easier comparison in Figure 2D and F?

Response:

Thank you for this suggestion. While CD4⁺ and CD8⁺ T cells were analyzed in separate staining panels due to technical constraints (CD45⁺CD4⁺CD25⁺Foxp3⁺ and CD45⁺CD3⁺CD8⁺IFN- γ ⁺ panels). Because both Foxp3 and IFN- γ were detected using PE, staining them in the same tube would lead to fluorescence overlap. Therefore, in our experiments, CD4⁺ and CD8⁺ T cells were stained and acquired in separate tubes with different antibody panels, and these populations were consequently not recorded within the same FCS file.

4. The results from subcutaneous and intratumoral injections, including tumor volume and survival, appear quite similar. Could the authors explain how these findings support their conclusion? Also, is there a method to quantify TP5 expression after

TP5-Ad injection?

Response:

We appreciate the reviewer's insightful questions. Regarding the similar antitumor efficacy between Ad5+TP5 and ADV-TP5, we have now clarified in the Discussion (lines 491-503) that both strategies ultimately deliver the same key components to the tumor microenvironment: replicative adenovirus backbone and pharmacologically effective levels of TP5. Under our dosing regimen, both approaches likely achieve sufficient TP5 concentrations for immune activation. The main advantage of ADV-TP5 lies in its ability to maintain this efficacy while simplifying TP5 delivery and improving translational feasibility by eliminating the need for separate TP5 injections.

For quantifying TP5 expression following ADV-TP5 injection, we used ELISA to detect the His tag fused to TP5. These results have been added as Figure S1F in the supplementary materials, demonstrating sustained TP5 expression in tumors after ADV-TP5 administration.

Figure S1. ADV-TP5 infects and replicates in murine tumour cells and expresses TP5.

5. The authors use the term ‘synergistically’. Please clarify how synergy was defined or calculated in this context.

Response:

We thank the reviewer for highlighting this important point. In this study, we used the term "synergistically" in a functional sense to describe the cooperative effects

observed at the level of tumor-associated macrophage (TAM) polarization and immune activation, rather than based on formal combination index calculations. Specifically, Ad5 and TP5 were considered to act synergistically when the combination treatment induced significantly greater promotion of M1-like TAMs and enhancement of CD8⁺ T cell function than either agent alone (as shown in Figures 2H and 6E). We have revised the description of the interaction between ADV and TP5 from “synergistic effects” to “TP5 enhances the effects of ADV” (line 24-25, 114-115, 282-284, 509-510, 518-519).

6. The manuscript refers to HSV, VV, and Ad5 showing similar effects. Please provide more detail about these viruses and clarify why similar responses were observed with three different oncolytic viruses.

Response:

Thank you for this suggestion. We have now provided additional details about the HSV and VV constructs used in our study in the Methods section (line 151-152). The similar responses observed across different oncolytic viruses (ADV, HSV, and VV) suggest that TP5's immunomodulatory effects are not virus-specific but rather represent a broad-spectrum enhancement of oncolytic virotherapy. This likely occurs because all three viruses induce immunogenic cell death and release tumor antigens, while TP5 provides complementary immune stimulation by promoting TAM repolarization and T cell activation. The experimental results for HSV and VV combinations are presented in Figure S4 (Line 443-445).

Figure S4. TP5 in combination with HSV and VV promotes infiltration of M1 macrophages and cytotoxic CD8⁺ T cells and enhances the antitumour efficacy of HSV and VV.

7. In Figure 2, IFN- γ ⁺ CD8⁺ T cells in the PBS and Ad5 groups are not significantly

different, but in Figure 6, a significant difference is reported. Please explain this discrepancy and re-check your raw flow cytometry data.

Response: We thank the reviewer for carefully noting this apparent discrepancy. We have re-examined all raw flow cytometry data, and found that there was a pasting error in the data in Figure 2 during the statistical analysis. After using the correct data for statistics, it was found that there was still no significant difference in IFN- γ ⁺ CD8⁺ T cells between the PBS group and the ADV group (P=0.0607), but this might be within an acceptable range of data fluctuations. We have updated Figure 2 (Line 700).

Figure 2. During oncolytic virotherapy, TP5 further increases the number of proinflammatory TAMs and cytotoxic T cells in the tumors of 4T1 tumor-bearing mice.

Minor Error:

In Figure 5H, it appears that groups with neither Ad5 nor TP5 s.c. injection are shown. Please remove this data from the figure to avoid confusion.

Response:

Thank you for catching this error. We have updated Figure 5H in the revised manuscript, removing the extraneous ADV and TP5 groups to maintain clarity and focus on the relevant experimental conditions.

Figure 5. ADV-TP5 suppresses tumor growth in solid tumor models through the regulation of CD8⁺ T cells and macrophages.

Reviewer #2 (Remarks to the Author): Authors combine adenovirus Ad5 with an immune regulatory peptide (thymopoietin 5, TP5) to enhance the immune rejection of tumors. They clearly demonstrate that the combination increases T cell infiltration and proinflammatory macrophages. Immune responses against the tumor are demonstrated by rejection of tumor rechallenges. In a second part, a recombinant virus (TP5-Ad) is constructed expressing GFP and E1a under the CMV promoter and TP5 (with signal peptide for secretion and 6His tag to be detected) under the EF1 promoter. Note that this virus is not tumor selective (E1a is under the CMV promoter, which is not tumor-selective) and that the mere transduction (entry) of this virus will induce GFP and TP5 expression (this virus works as a regular expression vector, where the transgenes are expressed without the need of any replication). It would have been much better to have a really oncolytic (tumor-selective) adenovirus.

Major issues

Mouse cells are not permissive to human adenoviruses. Authors should quantify levels of infection (transduction or entry of virus) and replication (production of progeny virus) in the three mouse cell lines used (4T1, CT26 and H22). Infectivity or transduction can be done in vitro infecting cells at different MOIs with reporter vectors (TP5-Ad for example) and counting the amount or percentage of transduced cells. Progeny production (burst size) can be measured after infection at different MOIs, washing, and incubating for several days to measure the PFUs produced.

In the experiments with the TP5-Ad (Fig 4) 4T1 cells infected at MOI=2 produce some TP5 secreted protein (Fig4B) which implies that some (unknown) level of transduction is achieved. Still, the level of transduction (percentage of transduced cells) at this MOI and other MOIs should be evaluated for the three models. GFP intensity (Fig4) does not quantify the transduction levels and E1a expression or cytotoxicity does not demonstrate viral replication (despite authors assume it). How many cells (percentage) are GFP+ at different MOIs? How much progeny virus is produced?

Response:

We appreciate this critical point. To address the permissiveness of mouse cells to human adenovirus, we conducted comprehensive experiments to quantify both viral transduction and replication. We infected 4T1, CT26, and H22 cells with ADV at different MOIs and measured the proportion of GFP-positive cells by flow cytometry

(Figure S1A-C). Additionally, we evaluated progeny virus production by measuring Hexon expression 72 hours post-infection (Figure S1D-E). These results demonstrate that while mouse cells show lower permissiveness compared to human cells, significant transduction and replication still occur, supporting the biological relevance of our mouse models for evaluating adenovirus-based therapies. We have made the corresponding changes in the text: Similarly, when tumour cells were infected with ADV-TP5 at increasing MOIs, the proportion of GFP⁺ cells rose in an MOI dependent manner, indicating that ADV-TP5 efficiently infects and replicates in tumour cells (Fig. S1A-C) (line359-362). We further evaluated the in vivo behavior of ADV-TP5 (Fig. S1D-G). Consistent with the results obtained in vitro, ADV-TP5 effectively infects and replicates within tumor tissues while expressing TP5 (line365-367).

Figure S1. ADV-TP5 infects and replicates in murine tumour cells and expresses TP5.

In a similar way, in the in vivo experiments should be an evidence that oncolysis (viral replication) is happening in the mouse tumors. This is needed to demonstrate that the therapeutic and tumor remodeling effect is associated to tumor cell infection and oncolysis. For this, the amount of viral infection (hexon staining) obtained after the intratumoral injections should be evaluated. Without such evidence one can assume that the inflammatory and tumor remodeling effect of the virus is due to the injected capsids, without any need of tumor cell infection. A control with inactivated virus (or empty capsids) should be included in the experiments.

Response:

This is an excellent suggestion. To demonstrate active viral replication in mouse tumors, we injected tumor-bearing mice with ADV-TP5 and collected tumor tissues at various time points (days 1, 3, 5, 7, and 9) to examine His and Hexon expression. The results (Figure S1F-G) show that ADV-TP5 replicates in mouse-derived tumor tissues, reaching peak levels on day 7.

Figure S1. ADV-TP5 infects and replicates in murine tumour cells and expresses TP5.

Furthermore, we administered inactivated ADV-TP5 to tumor-bearing mice and found complete loss of antitumor efficacy and immune activation (Figure S2), confirming that viral replication is essential for the observed therapeutic effects rather than mere capsid-mediated inflammation. We have made the corresponding changes in the text: To determine whether the antitumour effect of ADV-TP5 depends on viral replication, we used inactivated virus to exclude the contribution of viral capsid proteins.

Inactivated ADV-TP5 failed to exert antitumour activity (Fig. S2A-C) and likewise did not induce an effective antitumour immune response (Fig. S2D-G) (line380-383).

Figure S2. The antitumour efficacy of TP5 combined with ADV depends on replication-competent ADV.

Minor comments

Introduction

“Thymopentin II, a natural 49-amino acid polypeptide, is isolated from the thymus^{25,26}. TP5, an active component of thymosin II, is a pentapeptide corresponding to positions 32--36 of thymosin II 26-28.”

There seems to be an incoherence of thymopentin II and thymosin II names.

Response:

Thank you for identifying this terminology inconsistency. We have carefully reviewed and corrected the nomenclature throughout the manuscript, ensuring consistent use of "thymopentin" and related terms (line 92-95). All references now accurately reflect the established terminology in the field.

Results

Authors use a virus with CMV-EGFP-E1a and EF1-pA (empty cassette) as a control for their TP5-Ad virus. This virus should not be named “Ad5”.

Response:

We thank the reviewer for this clarification. We have standardized the viral nomenclature throughout the revised manuscript, using “ADV” consistently for the control virus and “ADV-TP5” for the recombinant construct, eliminating any potential confusion. We have revised the manuscript throughout, for example at lines 25,29,32,34 and so on.

Reviewer #3 (Remarks to the Author): This manuscript presents an innovative approach by combining thymopentin (TP5) with adenovirus 5 (Ad5) and evaluating its therapeutic efficacy across multiple tumor models. The authors have done an exceptional job characterizing the immune mechanisms underlying the antitumor effects of this combination, particularly in challenging models such as 4T1. The experimental design is rigorous, and appropriate controls were consistently included to distinguish the effects of combination therapy from monotherapies.

While the study is comprehensive and explores multiple aspects of the therapeutic strategy, several points would benefit from further clarification and refinement. The reviewer offers the following recommendations to enhance the manuscript's clarity and scientific depth:

1. Tumor Microenvironment (TME) Reprogramming: A more detailed explanation is needed regarding how the Ad5 + TP5 combination reprograms the TME in xenograft models. Including a discussion on this point would strengthen the mechanistic insight.

Response:

We thank the reviewer for this insightful comment. We have now expanded the Discussion to better explain how the Ad5 + TP5 combination reprograms the TME in our xenograft models, focusing on changes in the landscape of tumor-infiltrating lymphocytes (TILs). In brief, multiparameter flow cytometry profiling showed that Ad5 + TP5 (and TP5-Ad) increased the proportion and absolute number of activated, IFN- γ ⁺ CD8⁺ T cells, while concomitantly reducing immunosuppressive FOXP3⁺ regulatory T cells and lowering the CD8⁺ T cell/Treg ratio compared with either monotherapy. In addition, the combination treatment enriched effector-like and memory-like T cell subsets and enhanced expression of activation markers (CD25, CD69) within the TIL compartment. These coordinated changes in the TIL composition indicate that Ad5 + TP5 shifts the TME from an immunosuppressive, Treg-dominated state toward an effector-dominant, pro-inflammatory milieu, which provides mechanistic support for the superior antitumor activity observed in our xenograft models. This explanation has been added to the revised Discussion (line 483-490).

2. Flow Cytometry Gating Strategy: Although gating strategies are included, clarification is needed on how specific markers (e.g., CD8 vs. CD206, CD4 vs. F4/80 vs. CD69, CD3 vs. CD11b, and FoxP3 vs. IFN- γ) were distinguished when conjugated to the same fluorochromes. This is critical for reproducibility and interpretation of the immunophenotyping data.

Response:

We thank the reviewer for carefully examining our flow cytometry data. We apologize for the confusion. Although several antibodies are listed as being conjugated to the same fluorochromes (e.g., CD8 and CD206; CD4, F4/80 and CD69; CD3 and CD11b; FoxP3 and IFN- γ), these markers were not measured in the same staining panel. Instead, lymphoid and myeloid populations were analyzed in separate multicolor panels, and any markers sharing the same fluorochrome were assessed in different sample tubes/experiments, such that no single panel contained more than one antibody per fluorescence channel. Thus, there was no spectral overlap between these markers within a given panel, and the populations were distinguished based on non-overlapping fluorochrome assignments in each specific staining combination. We have now clarified this point in the revised Methods section to improve reproducibility and interpretation of the immunophenotyping data (line 202-203).

3. Mechanistic Depletion Studies: The manuscript shows that depletion of CD8 or CSF1R partially abrogated the combination efficacy. It is recommended to consider a dual depletion strategy (CD8 + CSF1R) to determine whether a more complete loss of therapeutic effect occurs, which could further delineate the mechanisms involved.

Response:

We thank the reviewer for this valuable suggestion. We have now performed the dual depletion experiment (CD8 + CSF1R) as recommended. Interestingly, simultaneous depletion of both CD8⁺ T cells and macrophages did not produce significantly greater impairment of therapeutic efficacy compared to individual depletions (Figure S3). This suggests that both cell types operate in a complementary rather than completely independent manner in mediating the antitumor effects, and that eliminating either population is sufficient to substantially disrupt the therapeutic mechanism (line 387-398).

Figure S3. Depletion of CD8⁺ T cells or macrophages markedly impairs the antitumour efficacy of ADV combined with TP5

4. Plain Language Summary: The current plain language summary requires revision to be more accessible to non-specialist audiences. Simplifying terminology and clearly explaining the study’s significance will improve its utility for broader readership.

Response: We thank the reviewer for this helpful suggestion. We have completely revised the Plain Language Summary (lines 49-62) to reduce technical terminology and more clearly communicate the study's significance for broader readership. The new summary emphasizes the clinical relevance of combining approved immunomodulators with oncolytic viruses and highlights the practical advantages of our engineered virus approach.

“Some cancers are hard to treat because they can hide from the body’s natural defenses. A promising new approach uses specially designed viruses that infect and kill cancer cells while also alerting the immune system.

In this study, we combined one of these cancer-fighting viruses with an immune-boosting drug called thymopentin (TP5), which is already used in patients. When tested in mice with tumors, the virus-drug combo drew more immune cells into the tumors and improved their ability to attack cancer.

To make treatment easier, we created an all-in-one virus, called ADV-TP5, that delivers TP5 directly inside tumors. This engineered virus controlled cancer growth as effectively as giving a combination of virus and TP5, in both regular mice and mice with human-like immune systems. TP5 also boosted the effect of other cancer-killing viruses. Together, these findings suggest that adding TP5 to virus-based therapies

could offer a simpler, more affordable way to enhance cancer immunotherapy, and potentially help more patients benefit from this type of treatment.”

5. Language and Formatting Edits: Minor but important language editing is needed throughout. Specific suggestions include (these are just few examples):

- Line 90 (32–36): Revise for clarity
- Line 71: Replace “good” with a more precise descriptor
- Define abbreviations at first use (e.g., HSV1 in line 79, HCC in line 130)
- Lines 101–103 in the Introduction may be better suited for the Discussion section

Response: We sincerely thank the reviewer for the thorough language review. We have carefully addressed all the specific examples mentioned and conducted a comprehensive linguistic revision throughout the manuscript. This includes correcting “32–36” for clarity on line 90, replacing “good” with more precise descriptors, defining all abbreviations at first use, and moving lines 101–103 to the Discussion section as suggested.

6. Ethical Approval Statement: The manuscript lacks mention of ethical approval for the use of human samples (e.g., PBMCs or patient-derived tumor specimens). This should be explicitly stated to ensure compliance with ethical research standards.

Response: Thank you for your suggestion. We have now added the ethical approval statement for human samples: “Human tissue collection and subsequent studies were approved by the ethics committee of Nanjing Drum Tower Hospital, and informed consent was obtained from all participants (Approval number: 2016-05-17)” (lines 137-140).

7. Because the work is so extensive, a graphical abstract or figure outlining the major concept would significantly help the readers.

Response:

We thank the reviewer for this excellent suggestion. We have created a comprehensive graphical abstract that illustrates the major concepts and mechanisms of our study, which is now included as Figure 8 in the revised manuscript (line 460). This visual summary will greatly enhance reader comprehension of our complex experimental findings and proposed mechanisms.

Figure 8. A schematic diagram showing the mechanism by which TP5 powers oncolytic virotherapy by orchestrating TAMs and CD8+T cell.

Overall, this is a well-executed and impactful study. Addressing the points above will further improve the clarity, rigor, and translational relevance of the work.

Reviewers' comments:

Reviewer #1 (Remarks to the Author):

The authors have addressed most of the concerns raised in the previous review. The revised manuscript demonstrates clear improvement. Overall, the revisions are satisfactory and have clarified the key points raised earlier. I have no additional major concerns.

Minor comment:

Figure 4C: Please add an x-axis label to enhance clarity.

Response:

Thank you for catching this error. We have updated Figure 4C in the revised manuscript by adding the MOI label to the x-axis, which improves the clarity and accuracy of the data presentation.

Reviewer #2 (Remarks to the Author):

None

Reviewer #3 (Remarks to the Author):

No additional comments. Authors have sufficiently addressed all of my concerns.